# Farm-Scale Crop Yield Prediction from Multi-Temporal Data Using Deep Hybrid Neural Networks

**Martin Engen** [1][iD]**, Erik Sandø** [1][iD]**, Benjamin Lucas Oscar Sjølander** [1][iD]**, Simon Arenberg** [2]**, Rashmi Gupta** [1][iD] **and Morten Goodwin** [1,*][iD]

1 Centre for Artificial Intelligence Research (CAIR), Department of ICT, Faculty of Engineering and Science, University of Agder, 4879 Grimstad, Norway; martin.engen@outlook.com (M.E.); esandoe@outlook.com (E.S.); benjamin@sjolander.no (B.L.O.S.); rashmi.gupta@uia.no (R.G.)
2 inFuture AS, Stortingsgaten 12, 0161 Oslo, Norway; simon.arenberg@infuture.no
* Correspondence: morten.goodwin@uia.no

**Abstract:** Farm-scale crop yield prediction is a natural development of sustainable agriculture, producing a rich amount of food without depleting and polluting environmental resources. Recent studies on crop yield production are limited to regional-scale predictions. The regional-scale crop yield predictions usually face challenges in capturing local yield variations based on farm management decisions and the condition of the field. For this research, we identified the need to create a large and reusable farm-scale crop yield production dataset, which could provide precise farm-scale ground-truth prediction targets. Therefore, we utilise multi-temporal data, such as Sentinel-2 satellite images, weather data, farm data, grain delivery data, and cadastre-specific data. We introduce a deep hybrid neural network model to train this multi-temporal data. This model combines the features of convolutional layers and recurrent neural networks to predict farm-scale crop yield production across Norway. The proposed model could efficiently make the target predictions with the mean absolute error of 76 kg per 1000 m². In conclusion, the reusable farm-scale multi-temporal crop yield dataset and the proposed novel model could meet the actual requirements for the prediction targets in this paper, providing further valuable insights for the research community.

**Keywords:** farm-scale crop yield prediction; deep learning; hybrid neural network; convolutional neural network; recurrent neural network; Sentinel-2 satellite remote sensing data

## 1. Introduction

In Norway, sustainable crop yield production depends on the agro-climatic conditions, the persistence of rainfall, soil quality, and other infrastructural development [1]. As the global population has been increasing, it is a significant challenge for farmers to produce increased quantities and better quality grains [2]. In this paper, we focus on exploring farm-scale crop yield prediction. We believe that it will provide valuable insights to the farmers in terms of knowing the particular type and quantity of crops in growing seasons based on geographical location and other environmental factors. In addition, it will improve food security and aid decision-making at various administrative levels.

In the past decade, machine learning has become an increasingly researched topic, used to predict and improve crop yield production worldwide [3]. Multiple studies have shown that county-scale crop yield prediction models are well suited for regional or national applications. However, limited studies have been reported on farm-scale yield prediction [4]. The primary reason is the limited ground-truth data available for farm-scale (i.e., kg per 1000 m²) production [4] due to the lack of funding needed for sustainable agriculture and collecting cost-intensive satellite images [5,6]. However, these impediments in the agriculture sector seem to fade. In Norway, detailed agricultural reports, including farm-scale statistics, have been made publicly available since 2017.

One can access the high-resolution satellite images provided by the European Union's Earth observation program, Copernicus.

In this paper, we identify the need to explore the use of satellite images from the Copernicus Sentinel-2 mission and the availability of other farm-scale digital crop production data. In addition, we propose and build a novel deep learning model to predict farm-scale crop yields throughout Norway. To accomplish this research study, we worked with a joint research project named KORNMO (KORNMO: https://prosjektbanken.for skningsradet.no/project/FORISS/309876). The principal aim of this project is to explore the potential deep learning-based models to predict the added value in the agriculture industry in terms of production optimisation, quality management, and sustainability.

### 1.1. Problem Statement and Hypotheses

We identified that the weather data are helpful in predicting farm-scale yield production, and the satellite data are beneficial to predict regional-scale yield production [4–7]. This research study explores the possibilities of building neural network models that can utilise satellite data for farm-scale yield prediction. To further concretise the problem, we define four hypotheses that this research study will test:

- **H1: Satellite images of farms and their surroundings can be used to accurately predict farm-scale crop yields.**
  The first hypothesis assumes that farm-scale crop yield prediction is possible given the availability of enough per-farm crop yield data in Norway and satellite images. We will then do a comparative analysis with the preliminary experiment using weather data to predict crop yield production. This is a prerequisite for the subsequent hypotheses, assuming that satellite data contains independent variables affecting crop yield.
- **H2: Accurate field boundaries along with satellite images increase crop yield accuracy significantly.**
  This hypothesis assumes that differences between field conditions and management decisions in neighbouring farms can effect crop yield, which could be difficult for a model to learn unless accurate field boundaries are provided. If the hypothesis is correct, it may show that satellite images can explain differences in crop yield between neighbouring farms and that such models can aid in decision-making at a farm level.
- **H3: Prediction accuracy can be further increased by combining satellite images and weather data.**
  It is assumed that weather data and satellite images contain some different and independent variables. We hypothesise that a deep learning model can learn features from both datasets effectively and that this provides better performance than models using the two data sources separately. Higher prediction accuracy makes the prediction model more helpful in aiding farmers and subsequent industries.
- **H4: It is possible to predict farm-scale crop yield earlier in the growing season with some reduced accuracy.**
  The most accurate crop yield predictions will likely be when there is as much data available as possible, meaning at the end of the growing season. However, getting accurate estimates for deliveries earlier allows mills and administrative authorities to prepare in advance.

Based on the proposed hypotheses, the authors investigate that the deep learning has emerged as the current state-of-the-art for crop yield prediction using remotely sensed data [8,9]. The progression of remote spectral observations for vegetation analysis over time was also investigated. From past decades, crop yield prediction has been a well-established research area and most research uses traditional statistical methods and handcrafted features derived manually from satellite images [9]. In recent years, automatic feature extraction from multispectral images using deep neural networks has outperformed conventional methods that rely on handcrafted features. Therefore, we address the state-of-the-art knowledge on crop yield prediction using satellite images, so-called remote sensing. This makes the alignment with the proposed hypotheses more apparent.

*1.2. State-of-the-Art Approaches to Crop Yield Prediction*

1.2.1. Origins and Early Use of Remote Spectral Observations

In 1973, NASA launched the world's first Earth-observing satellite named Landsat 1. The intent of Landsat 1 was to monitor and study the landmass of planet Earth. The satellite had two instruments to carry out the data collection: a primary camera system named the Return Beam Vidicon (RBV) and a secondary and experimental multispectral scanner (MSS) [10]. One significant project conducted by Rouse et al. in 1974 studied how the Landsat 1 MSS data could provide quantitative regional vegetative information of the farmlands throughout the Great Plains Corridor rangeland in America [11]. Later, in 1975, America launched the Large Area Crop Inventory Experiment (LACIE) in a joint effort between the United States Department of Agriculture (USDA), the National Oceanic and Atmospheric Administration (NOAA) of the Department of Commerce, and the National Aeronautics and Space Administration (NASA). This project aimed to evaluate and prove the economic importance of applications built using remote sensing from space. The LACIE project concentrated on wheat grown in North America and combined Landsat data with NOAA's meteorological information to run experimental investigations on the crops. The ultimate aim was to satisfy the requirements of being able to monitor and make crop production inventories on a global scale [12].

In 1976, the Agricultural Research Service (ARS) of the USDA continued developing agrometeorological models for forecasting wheat yields [13]. Even though the use of remote spectral observations for forecasting models received scepticism at the time, earlier research by [11–13] demonstrated that it would be both of value and technically feasible. The spectral observations were to be collected through handheld, aircraft, or spacecraft-mounted sensors to combine these observations with soil property and daily increments of weather data to estimate the yield of the saleable plant parts ultimately. Experiments showed spectral observations could calculate vegetation indices. Which could measure the amount of green photosynthetically active tissues and estimate reliably the leaf area index (LAI), both of which could be used as input to their models [13]. As proven by [11–13], the multispectral images carry information that is useful for crop yield-related experiments, but it is challenging to extract the full potential of the data. The most promising avenue at the time was the use of vegetation indices.

1.2.2. Deriving Values from Vegetation Indices

After the initial research and experiments on vegetation and yields using MSS data collected by Landsat, there was further work to continue these efforts. One common and central theme seems to revolve around the use of vegetation indices [14]. Vegetation indices can capture the information of multispectral images into a format well suited for experimentation using existing models and techniques. Bendetti et al. conducted a study to investigate the potential use of NDVI applied to spectral imagery collected by NOAA satellites in Italy in the period 1986 to 1989 [15]. The study considered the production of wheat of the provinces in the region Emilia Romagna. The spatial resolution of the images collected was about $1 \times 1$ km and was applied to test sites of 900 ha ($3 \times 3$ km), resulting in each test site having $3 \times 3$ pixels of data per image. The researchers calculated the NDVI of each pixel and calculated the mean of these represented the real NDVI that provided reasonable yield estimates at a province and regional scale.

In 2014, Johnson assessed the use of remotely sensed variables for forecasting corn and soybean yields in the United States. In this case, the remotely sensed variables were satellite multispectral images collected from Terra satellites, daytime and night time Land Surface Temperature (LST), and precipitation. The satellite images were masked based on the Cropland Data Layer (CDL) produced by the NASS, such that only pixels connected to soybean or corn remained, and an NDVI was calculated for each pixel. Much like Bendetti et al., the representative NDVI of each day was the mean of all NDVI values from an image [14]. Thus, we identify that vegetation indices have been proven to work adequately with crop yield prediction and estimation. Given that satellites had a relatively

poor resolution for many years, using vegetation indices such as NDVI allowed it to capture vegetational relevant properties into a single feature that is well suited for various linear models.

### 1.2.3. Machine Learning Applied to Remotely Sensed Data

There has been increasing use of machine learning and deep learning techniques on remotely sensed data to estimate and predict different crop yields [9,16]. Various studies have found that non-linear approaches outperform linear models when predicting and estimating yield with remotely sensed data [14,16,17]. Crop phenology (phenology is defined as "a study of the timing of recurring biological events" [18]) varies from season to season, depending on environmental and managerial factors [19]. Jiang et al. set out to explore a phenology-based LSTM model for corn yield estimation [17]. The corn crop includes six distinct phases of development throughout the season: planted, emerged, silking, dough, dent, and mature. Jiang et al. split these into five Growth Phases (GP) (the growth phases as identified by Jiang et al. GP1: planted to emerged; GP2: emerged to silking; GP3: silking to dough; GP4: dough to dented; GP5: dented to mature), where one growth phase symbolises one time step for the LSTM. Each time step included three meteorology features and a single vegetation index WDRI (Wide Dynamic Range Vegetation Index (WDRVI) is a vegetation index similar to NDVI. However, it will be less affected by the saturation effect when the density of biomass is high [17]), in total 4 features $\times$ 5 time steps, with this, the LSTM should estimate county-level corn yield. With ten years of training data (2006–2015), they saw an RMSE of 0.87 metric tons per hectare [17]. This result is better than what [14] could manage using the RuleQuest Cubist (0.96), but cannot be directly compared due to the number of and which seasons involved in the training is not the same.

In 2017, You et al. developed a novel approach by combining convolutional and LSTM networks. The researchers predict the soybean yield on a county-level scale in the USA using multispectral images [9]. As far as we know, their research is the first to use the raw images as input to the deep learning algorithms. You et al. argue that using handcrafted features such as NDVI can be pretty crude. Using deep learning to find the relevant features in multispectral images automatically can be more effective. Both convolutional and LSTM networks were trained on extracted features. They saw that these networks significantly outperformed competing methods, such as ridge regression, decision trees, and DNNs, with an RMSE reduction of 30% compared to the best of the competing models. You et al. demonstrate that deep learning models can automatically find relevant features for yield prediction from multispectral imagery and that handcrafted features might not be necessary [9]. Inspired by You et al. [9] and their histogram approach, Sharma et al. trained neural networks using raw satellite images into a CNN-LSTM model. Their approach forgoes any handcrafted or rudimentary features, such as vegetation indices or histograms. Instead, they use a convolutional neural network to perform all necessary feature extraction and learn the best representation for yield prediction. They argue that prior work has not considered surrounding factors, such as water bodies or urban areas that may effect crop yield [8]. By comparing the proposed models with the histogram approach of You et al. [9] and other machine learning and regression methods, they show that using raw satellite images outperforms all previous methods on these data. In addition, they show that adding land cover masks improved their model by 17%, which suggests that contextual information is essential for these models.

Khaki et al. in 2019 [20], proposed a CNN–RNN framework to predict crop yield production for corn and soybean yields in the United States for years 2016, 2017, and 2018 using historical data. The researchers implemented various deep learning methods such as random forest, deep fully connected neural networks, and LASSO and did comparative analysis with the newly proposed model. They found RNN-CNN model achieved a root-mean-square-error 9% for corn and 8% for soybean, which outperformed all other implemented methods. Later, in 2021 [21], the researchers used expanded data collected

from more counties across the United States, i.e., covered 1132 counties for corn and 1076 counties for soybean. They proposed a new convolutional neural network model called YieldNet, which utilised transfer learning between corn and soybean yield predictions by sharing the weights of the backbone feature extractor. In addition, to consider the multi-target response variable, they proposed a new loss function, which could make accurate early predictions before the harvesting period.

A recent study by Sagan et al. in 2021 presented the use of raw satellite images for field-scale level yield prediction. Although You et al. used raw satellite image data, they also condensed it using pixel value histograms. As well as to investigate the use of raw images, Sagan et al. also did experiments based on several handcrafted features, such as vegetation indices [22]. The researchers used the images collected from each plot for two main directions in their study: (1) condense them into handcrafted vegetation indices, and (2) use the raw images directly in a CNN-based model. Their results show that raw image-based deep learning performance was comparable, if not superior, to deep learning methods using handcrafted features. Overall, the root-mean-square error was about 10% regardless of crop-type and irrigation conditions. Their work showcase that an image-based deep learning approach can utilise spectral, spatial, and temporal information from the satellite data and essentially reduce the need for feature engineering. Most studies have evaluated the use of remotely sensed data on a county or regional scale, as discussed in this section. As crop yield statistics required to make predictions based on farm or field-scale have not commonly been available to the public [4]. Sagan et al. [22] made an effort to predict field-scale crop yields using deep learning by building a dataset comprising small experimental plots and making yield predictions for these plots. The results show the models can learn growth-related features, even on such small plots. In conclusion, this indicates that remote sensing with deep learning can be effective for field and farm-scale predictions, given that enough crop yield statistics are available.

We present our contributions thus: (i) an accurate farm-scale crop yield prediction using satellite images; (ii) identifying field boundaries by applying image masks on satellite images to improve the crop yield accuracy; (iii) fusing satellite image data and weather data for further crop yield prediction improvement; and (iv) predicting farm-scale crop yield production earlier in the growing season. In this section, we presented the overview of this research study and motivated the idea behind this study. In addition, we define the current state-of-the-art approaches to the crop yield predictions using remotely sensed data. In Section 2, we report on the history of Norwegian agriculture and grain production. We also report the relevant studies of using remote sensing data and building deep learning models in agriculture improvements. Section 3 presents the methodology of this research study. We will discuss the implementation of baseline and novel models to carry out the yield prediction in addition. Section 4 showcases the experiments conducted on the satellite images and compares the implemented models' performances. We summarise our results and assess them against our problem statement and the four hypotheses in Section 5. We conclude this research study and discuss further directions for future work in Section 6.

## 2. Related Work

### 2.1. Norwegian Agriculture and Grain Production

Norwegian agriculture has traditionally been family farming [23]. With the support of society and politicians, the goal is to reach national self-sufficiency based on the available natural resources. The government has designed the subsidy rates to compensate for any disadvantages to keep agriculture active and profitable. For example, land payments are differentiated by geography and type of agricultural production [24]. Although Norwegian agriculture has comprised smaller family farms spread out, there has been a decline in the number of farm holdings by 50% in the last three decades. Simultaneously, there has been an increase in the average size of farms, from 14.7 hectares in 1999 to 24.7 hectares in 2018 [25]. The most produced categories of food throughout the country include milk and milk products, meat, poultry, eggs, potatoes, and grains [23]. The Norwegian topography

consists mainly of mountain masses, and as a result, only 3% of the total landmass is cultivated land (excluding Svalbard and Jan Mayen). Because of differences in climatic conditions, a minor part of this cultivated land can grow cereal for human consumption [25]. Nibio has constructed a map by using climate divisions to showcase where the different climatic zones are located [25]. As per the research, Norway's eastern and southeastern part is best suited to produce food-grade grains. There are mainly four types of grain produced in Norway: wheat, barley, oat, and rye (and rye–wheat).

Barley is the most grown grain in Norway because of the need for a shorter growing period. The most suitable growing areas are in the north and in higher altitudes, where barley is used for animal fodder [26]. Wheat requires a longer growing season compared to barley. Wheat can be harvested in two categories, i.e., spring wheat and winter wheat. Winter wheat resumes growing in the spring and is harvested in the summer. The result is the same type of grain, but winter wheat may give higher yields as it will resume growing earlier in the spring compared to spring wheat [27]. Oats thrive in cold and moist climates, which makes them well suited for cultivation in Norway. The vast majority, i.e., over 90% of oats grown, are used for animal feed, and 2% are used for human consumption [28]. Rye is the least grown grain in Norway, covering 2% of the total area used for grains [29]. Rye thrives in higher altitudes, but is mostly only grown in the east and southeastern parts of the country [29,30].

*2.2. Plant Growth Factors*

Plant growth and wellness are affected by elements from its surroundings. According to Woodward F. Ian [31] and Oregon State University [32], the four main environmental factors affecting plant growth are light, temperature, water, and nutrition.

- **Light:** Light is a component of photosynthesis and is essential for overall plant growth. In Norwegian crops, the duration of light is particularly relevant; according to Åssveen and Abrahamsen, the duration of light in a day (day length) is more influential than temperature as growth factors [33].
- **Temperature:** temperature affects growth in several ways. A rise in temperature triggers the germination process, so the temperature controls when the seedlings initially sprout [34]. The temperature also affects when crops such as winter wheat break dormancy to resume the growth in spring [35]. A typical measurement using temperature to estimate plant growth is the so-called sum degrees, meaning the sum of mean daily temperatures for the period. Crops will have different requirements for how much sum degrees are needed before it is ripe for harvest [31,32].
- **Water:** together with light, water is a primary component of photosynthesis, and consequently, an essential factor for growth. For crops, water can come in the form of direct precipitation, humidity, or irrigation [31,32].
- **Nutrition:** plants need in total 17 essential chemical elements to grow. Three of the required components are found in air and water (carbon, hydrogen, and oxygen), while the soil must provide the rest [36]. Farmers can fertilise the soil, which adds materials containing nutrients to make these available to the plants. The roots absorb approximately 98% of the nutrients through soil water [37]. If the plant is under stress by extreme temperatures, drought, or low light, this can lower the plants' ability to absorb nutrients efficiently [38].

With the understanding of Norwegian agriculture, grain production and plant growth factors, we are now exploring deep learning models to predict crop yield production. We will report on remote sensing techniques used by researchers in predicting crop yields and how it helps analyse vegetation indices.

*2.3. Deep Learning Models in Crop Yield Prediction*

In crop yield prediction, the researchers used a supervised algorithm called perceptron [39]. There are two types of perceptrons, i.e., single layer perceptron and multilayer perceptron. Single layer perceptron algorithm is a type of binary classification which en-

ables the artificial neurons to learn and processes elements in the training set one at a time automatically [40,41]. Multilayer Perceptron (MLP) is constructed by combining multiple perceptrons and placing them in layers [41]. An MLP architecture can be considered a *Deep Neural Network* (DNN) when the stack of hidden layers is big enough, although the exact number of layers required for it to be considered *deep* is not clearly defined [40].

### 2.3.1. Convolutional Neural Networks

In 1981, inspired by research on the receptive fields in the visual cortex of cats and monkeys, Kunihiko Fukushima created a new layered hierarchical architecture which he called the *neocognitron* [42]. Convolutional Neural Networks (CNNs) share the same basic architecture as the neocognitron. CNN uses a combination of convolutional layers, and sub-sampling layers called pooling layers. They are widely used in the field of computer vision, taking images represented as three-dimensional matrices of pixels (width × height × channels) as input [43]. Both convolutional and pooling layers take a 3D matrix as input and outputs a new 3D matrix, typically with fewer pixels than the previous layer. In deep neural networks, each neuron is directly connected to all the neurons in the previous layer, allowing each to learn global patterns across its input space. This approach enables fully connected layers to learn complex patterns on the input space. As a result, it limits the ability to detect local correlations at any position in the input space, but it can identify a local pattern in any different position of the input space by simply adding enough neurons [44]. This leads to an inefficient network architecture, which increases computational costs and requires large datasets that include pattern samples in all possible locations.

### 2.3.2. Recurrent Neural Networks

Recurrent Neural Networks (RNNs) are a family of neural networks that take data in sequences as input. Recurrent neural networks are known for their ability to "*remember an encoded representation of its past*" [8] by passing on the output of the previous time step along with the input at the current time step. Each time step of the sequence is processed using the same weights as all other time steps, significantly reducing the number of neurons required to process long data sequences. Reusing the same weights for each time step allows RNN to generalise well even on sequences of varying lengths, as the output can be extracted from any calculation step. A known limitation of simple RNNs is their inability to retain information across long sequences of data because of the vanishing-gradient problem that arises with very deep neural networks [45,46]. To solve this issue, Hochreiter and Schmidhuber released a significantly more complex RNN cell called the Long Short-Term Memory (LSTM) cell in 1997 [47]. Although LSTMs have proven superior to standard RNNs, they are also more complex and require more computations to train [47]. A newer variant of LSTM called the Gated Recurrent Unit (GRU) was simpler to compute and implement [48]. The primary purpose of GRU is to drop information that is no longer relevant and to control how much information is carried over to the next step, which helps the RNN to remember long-term information [48].

### 2.4. Remote Sensing Technique in Crop Yield Prediction

Remote sensing is defined as "the field of study associated with extracting information about an object without coming into physical contact with it" [49]. Although the definition is broad, the term is primarily used in the context of earth observations using optical imaging instruments on-board satellites or aircraft [49,50]. Satellites used for earth observations often carry a unique optical imaging sensor, making them capable of measuring the earth's reflectance in multiple spectrums. For example, the Sentinel-2 satellites carry a Multi-Spectral Instrument (MSI) measuring the reflectance of the earth in 13 spectral bands, from the Visible and Near-Infrared (VNIR) to Short-Wavelength Infrared (SWIR) [51]. The images produced by such instruments allow us to look at unknown parts of the earth and vegetation indices that are simply invisible to human eyes. In addition, it captures the

changes in plant growth over time that can span anything from minutes to decades of measurements [49].

Identifying Vegetation Index

A vegetation index is a type of feature engineering in which several spectral bands are combined to form compact and more manageable vegetation features. A widely used type of vegetation index is the Normalised Difference Vegetation Index (NDVI). The NDVI uses known properties of vegetation and their reflectance to show whether a pixel from a multispectral image contains healthy vegetation and to which degree [52]. The specific relevant properties used to calculate the NDVI are:

1. The leaves of plants contain chlorophyll pigments, which is an essential factor in photosynthesis and ultimately makes the leaves green. Chlorophyll pigments make the leaves absorb a lot of the red and blue regions of the visible and near-infrared (VNIR) spectrum, but not in the green region. The number of chlorophyll pigments can indicate health in vegetation; thus, measuring the amount of reflection in the red spectrum can estimate vegetation health. Low reflectance in the red spectrum shows healthy vegetation.

2. Leaves have evolved to scatter solar radiation in the Near-Infrared (NIR) part of the spectrum, as it is difficult to extract the energy at these wavelengths efficiently (longer than 700 nm). This implies that healthy vegetation will have higher reflectance of NIR.

Based on these known properties, the formulae for calculating the NDVI for any given pixel in a multispectral image can be seen in Equation (1). The pixel values from the near-infrared (NIR) and red (R) bands are used to calculate the NDVI for that specific pixel location. One method to utilise the NDVI values for existing models and frameworks is to get the NDVI of all relevant pixels and do an arithmetic mean to get a single value representing the NDVI of the entire area [14,15].

$$NDVI = \frac{NIR - R}{NIR + R} \tag{1}$$

## 3. Material and Methods

In this section, we present the research methodology for this study. As shown in Figure 1, we discuss the sources of the collected multi-temporal data and the techniques to handle these data in Section 3.1. We present the data pre-processing techniques in Section 3.2. In addition, we present the feature extraction techniques used to expand the dataset and combat overfitting. We then present the proposed prediction models to predict per farm-based crop yield production in three subsections based on processed data: Section 3.3 includes a baseline weather data model; Section 3.4 introduces two initial experiments focusing on single and multi-temporal satellite images; and Section 3.5 presents three newly proposed approaches using a combination of weather data and satellite images.

### 3.1. Multi-Temporal Data Collection and Data Handling

There is no readily available single dataset that can be downloaded and used for machine learning in yield prediction in Norwegian agriculture. Therefore, a large portion of this work has been collected from different sources that can be used for crop yield prediction on a farm-scale level. This section explains which data sources are used and how the data have been collected.

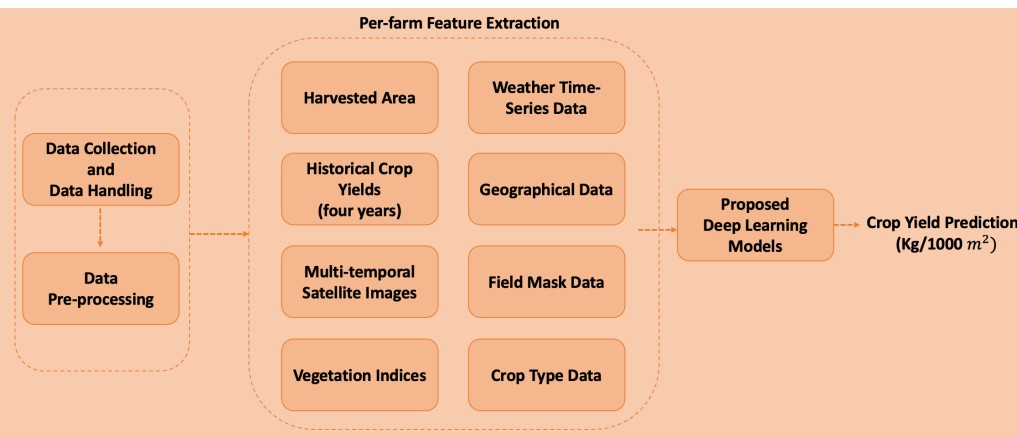

**Figure 1.** Research methodology for this research study.

### 3.1.1. Norwegian Agriculture Agency

The primary data sources for the project are the official public archives of farmer grant applications and grain deliveries from the Norwegian Agriculture Agency. As Norwegian grain farmers rely on subsidies, they must fill out yearly grant applications describing the land used for crop cultivation. Some of these data are publicly available. These are used to build a base dataset that can integrate other data sources through each farmer's unique organisation number (all farmers are registered in the official registers (*Brønnøysundregistrene*), giving them a unique organisation number). From the Norwegian Agriculture Agency, three different yearly reports serve as the base data for this research study. First, the grain delivery reports that include how much grain of different types, i.e., barley, oats, wheat, rye, and rye wheat, each farmer has sold in the last year. Second, the agriculture production subsidies from farmers regarding the area of cultivated land. These areas provide the relative crop yield for each farm and crop type from the year 2017 to 2019 for crop yield prediction [53]. The area of cultivated land use is the only missing information that prevents data from 2012, through 2016, from being used in our experiments. Third, As farmers must submit which land areas are used, the Norwegian Agriculture Agency has detailed reports linking farmers' organisation numbers to all used cadastral units (an area of land, as specified in the official Norwegian cadastre (*matrikkelen*)). We served these data as a basis for precise geographic location and masking of images, as will be described in forthcoming sections.

### 3.1.2. Geographical Data

Precise geographical mapping for each farm is required to retrieve accurate remote sensing data. The Norwegian Institute for Bio-economy Research (NIBIO) provides a map of cultivated land areas throughout Norway. Together with cadastral identifiers from the land use data, these two map layers allow us to create precise geographical mappings for each farm in the dataset. Figure 2a,b show a visualisation of the two map layers, further explained in this section.

The Norwegian cadastre contains millions of geographical entities, each one is describing some land area with a unique label or identifier. The cadastral identifiers are composed of a commune number and three numbers describing the exact property and section. Using the land-use reports from the Norwegian Agriculture Agency, a collection of geographical shapes for all farms, each year from 2017 to 2019, was extracted from the cadastre. However, the cadastre lacks information about agricultural land; the geographical data do not separate forested areas, water, and other land types from cultivated land (visible in Figure 2a). New and updated cadastral identifiers for old values are obtainable through a public API at Geonorge (https://ws.geonorge.no/kommunereform/v1/), which was used to create a mapping from the old to new identifiers compatible with the latest cadastre.

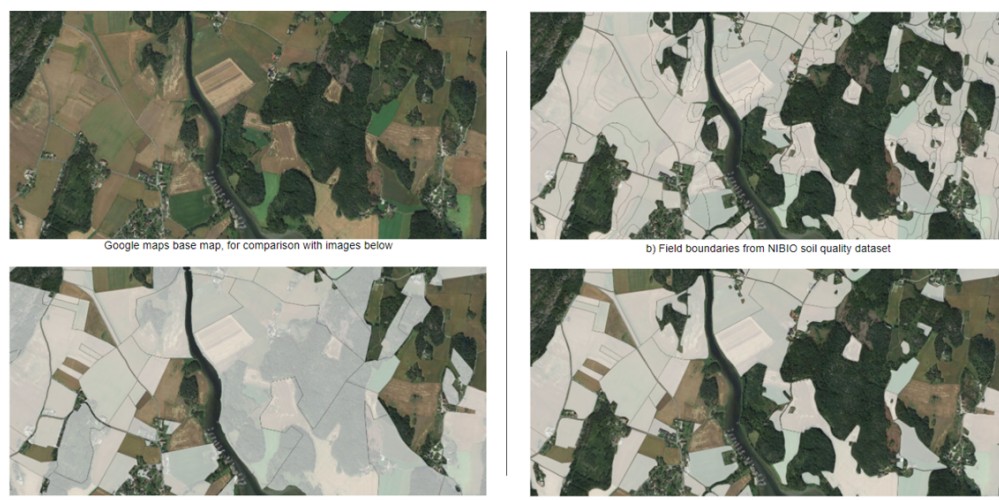

**Figure 2.** Visualisation of the geographical layers, overlaid on a satellite images.

We further track soil and field-related features, such as quality, organic material, and water storage capacity, throughout Norway. These datasets are made publicly available by Geonorge (https://www.geonorge.no/) and can be downloaded in formats such as GDB (geodatabase), which making it possible to distinguish fields from the rest of the environment (such as lakes, forests, and towns). A well-suited dataset for this is the soil quality (in Norwegian: *jordsmonn*: https://kartkatalog.geonorge.no/metadata/jords monn-organisk-materiale/6898f450-01ea-4b1c-b284-194308de1445) dataset, illustrated in Figure 2b, which, according to NIBIO, maps roughly 50% of all cultivated fields in Norway. The soil-quality dataset includes detailed information on the field boundaries and further classification of soil quality within these fields. For this project, only the field boundaries are used. A new geospatial dataset is created by extracting the intersection between the two layers while keeping cadastral attributes to connect field boundaries from NIBIO with the cadastral boundaries data from the cadastre. The output dataset has precise field boundaries for each farm. See Figure 2c for illustrations of the two base layers (cadastral and field boundaries) and the resulting intersection.

### 3.1.3. Weather Data

The weather is one of the main external factors that are crucial for farming [54]. Grain farmers depend on periods with little precipitation in the spring so that the fields are dry enough to support heavy equipment for ploughing, harrowing, and sowing. After sowing, the temperature must be stable so that the seedlings sprout and precipitation throughout the summer is required to water the plants [55]. As the grains mature, a period of limited precipitation is needed, so that a combined harvester can harvest the grains before they can be delivered to the mills. As shown by [14], temperature is highly correlated to the eventual yield, and precipitation is relevant when there is no irrigation used [56]. The Norwegian Meteorological Institute (MET Norway) collects weather data across Norway and makes it publicly available through the Frost API (https://frost.met.no/). MET Norway has 1578 weather stations throughout Norway, where roughly 840 includes temperature and 630 includes precipitation data, with some variations from year to year. The temperature measurements are available at one-hour intervals, but that level of granularity is not required for this project. It split temperature measurements into individual days, where a min, max, and arithmetic mean are stored. The accumulated precipitation is downloaded from each weather station with these measurements, resulting in one precipitation feature per day.

Previously, the weather at each farm was estimated using the readings from its nearest weather station [53]. This approach has been further improved by including weather stations with intermittent data. Interpolation (interpolation using linear triangulation

was attempted but was dismissed because too many farms are outside the bounds of the possible triangulation arrangements) using neural networks is used to estimate the weather at each farm to reduce this difference. By training two deep neural networks to create soft sensors, lower deviations are achieved than nearest neighbour interpolation. Training samples are designed by keeping the sensor's reading as the actual/desired output value and providing the readings of the three closest sensors and their normalised latitudinal, longitudinal, and vertical (distance from sea level) differences as inputs. The trained model is then used to create soft sensors at the location of each farm. Temperature and precipitation models are trained separately since the temperature and precipitation sensors often have different geographical locations. A deeper network showed a slightly lower prediction error for the temperature model than the precipitation model, where additional depth provided no significant benefit. As shown in Table 1, when compared to the nearest neighbour, the DNN models (a dense neural network built for the preliminary project [53] further improved the predictions compared to the results reported in Table 1 by including weather stations with intermittent data) achieve a 23% reduction in mean absolute error in soft precipitation sensors and a 67% reduction in mean absolute error in soft temperature sensors.

**Table 1.** Mean absolute errors in weather interpolation.

|  | Nearest Neighbour | Deep Neural Network | Change |
|---|---|---|---|
| **Precipitation error** | 1.5 mm | 1.15 mm | $-0.45$ mm ($-23\%$) |
| **Temperature error** | 1.6 °C | 0.52 °C | $-1.08$ °C ($-67\%$) |

### 3.1.4. Satellite Image Data

Using satellite images is the current state-of-the-art for farm-scale crop yield prediction with no intrusive and labour-intensive monitoring, as described in [8,9,22]. Therefore, it is also a significant focus of this research study. Building a dataset of multispectral satellite images for farm-scale crop yield predictions relies on the availability of high-resolution satellite images combined with precise geographical information about farms [22]. Therefore, the Copernicus Sentinel-2 satellite mission is the source of all images used in this research study. The Sentinel-2 mission is developed by, and operated by, the European Space Agency. It comprises two polar-orbiting satellites (S2A and S2B) that provide high-resolution images of the earth every five days at the equator and more frequent at higher latitudes (https://sentinel.esa.int/web/sentinel/missions/Sentinel-2). Images from these satellites were accessed through Sentinel Hub, a subscription-based cloud API for satellite imagery (https://www.sentinel-hub.com/). The Sentinel-2 satellites carry a multispectral instrument that captures optical images in 13 spectral bands, including visible light (red, green, and blue channels), NIR, and SWIR.

Using the geographical shape files described earlier in this section, each farm's shape is converted into a point at the geometry's centroid, which is then used to build a $2 \times 2$ km bounding box. With $2 \times 2$ km bounding boxes, roughly 65% of all farms in the dataset are >90% found covered. This size provides good coverage of farms in the dataset while not too large, leading to a larger image size or reduced resolution (in meters per pixel). In preparation for the experiments, a dataset of 509,910 unique Sentinel-2 images was pre-processed. The resolution of the images is $100 \times 100$ pixels, meaning a single-pixel roughly represents an area of $20 \times 20$ square meters. The cloud API handles up-scaling the 10 m resolution bands and down-scaling the 60 m resolution bands, using nearest neighbour interpolation. All 12 channels were of equal size. Temporal changes for each farm in the dataset are captured by having multiple images of the same farm throughout the growing season. As a result, 30 images (from 1 March to 1 October of each year) are downloaded per farm, with a mean temporal resolution of 7 days for which the best image is queried, based on the least cloud coverage (selecting the least cloudy image for every

7 days is handled by the Sentinel Hub API). The result is an image time series for each farm, with weekly images from approximately week 10 to 39 (see Figure 3).

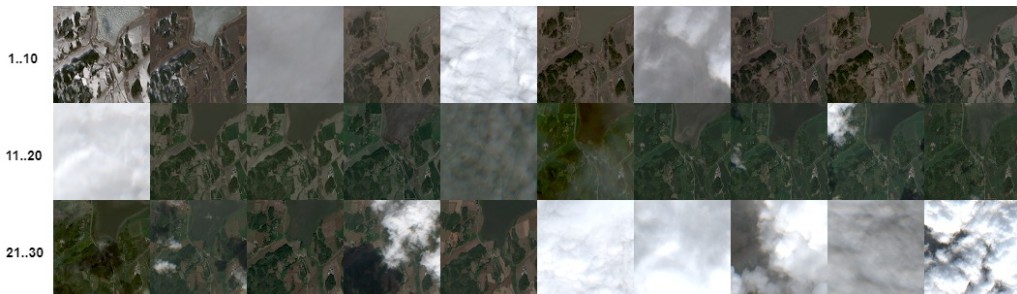

**Figure 3.** Dataset sample showing a 30 week time series of a farm in natural colour. Images are indexed 1–30, roughly equal to weeks 10–39.

### 3.1.5. Data Handling Using Image Masking

Image masking makes it possible to focus on portions of an image that is of interest. Masks can be applied so that it either highlights certain parts or removes irrelevant parts from the image [57]. In this research study, provided we know where the cultivated fields are located, masks can be used on the satellite images to remove everything not registered as a cultivated field or as extra information to highlight where the fields are located [58].

1. **Generating masks:** the image masks are generated in three steps:
    - Intersect the bounding box and the cultivated fields for each farm, leaving only the coordinates for cultivated fields inside the satellite images.
    - Convert the geographic map coordinates of longitude and latitude to corresponding pixel locations of the satellite images. Given that the images are $100 \times 100$, the bounding boxes represent the borders of these images (top left 0,0 and bottom right 100,100). Convert each field point within the bounding box to pixel coordinates based on relative position within the bounding box.
    - Generate a matrix of zeros and ones based on where the cultivated fields are located, resulting in a $100 \times 100$ matrix.

2. **Applying masks:** we implemented two methods of applying masks to the images; applied or added as a channel. In the first method, i.e., applied mask, the mask is multiplied by each image channel. The result is an image where only the cultivated fields remain, and all other pixel values are zero. In the second method, i.e., added a mask, the mask is added to the image as a separate channel. The mask channel will add some information to the image so that it can highlight which parts of the image are cultivated fields.

### 3.1.6. Time Spans of Satellite and Weather Data

There are two time series in these dataset: satellite images and weather data. In Norway, the growing season for grain usually starts mid-April, and harvesting is usually done in August. However, it can also occur from July until late September, depending on seasonal variations. Between 1 March and 1 October is downloaded for both satellite and weather data to encapsulate the entire growing season within the dataset.

In conclusion, the dataset for the upcoming experiments through all the sources mentioned above is utilised. It contains the following features per farm per year for the last three complete production years (2017–2019) (to clarify, the primary key for each sample is the farm-year combination. This gives us three samples for a single farm that has produced grain for all three years): daily min, mean, and max temperatures; daily total precipitation; latitude; elevation; weekly twelve-band satellite images; field mask; crop yields of the four previous years; harvested area; crop type; and relative crop yields.

*3.2. Data Preprocessing*

3.2.1. Normalisation

As we can observe from the previous section, we acquired data from various public resources, which must be normalised before training. Most features are normalised to fit a range between 0–1 better, using the linear scaling shown in Equation (2). The upper and lower values are the minimum and maximum features for many features, resulting in a min–max normalisation. For other features, such as the weather features, a fixed normalisation range is applied to keep the normalisation consistent across the entire time series and measurement aggregations (the temperatures for each day are aggregated by min, mean, and max). Table 2 shows the upper and lower normalisation values used in Equation (2) for all features that are not min–max normalised. The Sentinel-2 images have pixel values between 0 and 1 from the source and are therefore not normalised before use.

$$\text{normalised} = \frac{\text{value} - \text{lower}}{\text{upper} - \text{lower}} \tag{2}$$

Equation (2) shows how features are normalised. Lower and upper values are either specified in Table 2 or they are set to a feature's minimum and maximum values. For lower and upper equal to the minimum and maximum values, this is called min–max normalisation.

**Table 2.** The normalisation constants (lower and upper) used to scale feature values where min–max normalisation was not used.

| Feature | Lower | Upper |
| --- | --- | --- |
| Crop yield (kg/1000 m$^2$) | 0 | 1000 |
| Temperature (°C) | −30 | 30 |
| Precipitation (mm) | 0 | 10 |
| Historical yield (kg) | 0 | 10,000 |

3.2.2. Implementing Data Augmentation to Reduce Overfitting

The models proposed in this study are deep learning-based, which can be considerably data-hungry. The dataset involved spans three years (2017, 2018, and 2019) and comprises 509,910 unique images, whereas one sample contains 30 images, resulting in 16,997 samples in total. Overfitting could be observed in the proposed models that train on raw satellite images. Therefore, we extensively used data augmentation techniques on the images to increase the overall dataset size and combat overfitting. We implement three main data augmentation techniques: image cropping, image rotation, and random pixel noise. Because the memory requirements of the complete dataset are too large to fit in GPU memory or even RAM, images are continuously read from storage. Because both rotation and noise are performed with some randomness, a complete cycle of the augmented dataset is never the same. No data augmentation was performed on the validation samples. However, for models that take cropped images, only the centre crop was used for validation.

- **Cropping:** the cropping augmentation is a method to extend the dataset. Initially, the images are 100 × 100 pixels, and by cropping these to 90 × 90 pixels, we extend the dataset by a factor of five, with minimal loss of information. Each training sample is cropped five times, such that the resulting dataset has five entries for the same farm. The crops are done top-left, top-right, centre, bottom-left, and bottom-right.
- **Rotation:** another standard method of increasing the number of training samples is to apply image rotation, which forces the models to learn features not purely based on the location of certain specific patterns in an image. We extensively apply image rotation to all image samples by selecting a random rotation angle for each image. The rotation angle is also different for images in the same time series, which produces a unique image time series every time. Image dimensions are kept unchanged, meaning that any corners rotated out of the image are discarded. Black pixels are used to pad any empty corners of the image.

- **Pixel noise:** to increase the variability of images, we introduce some augmentation through the noise, using a simple salt-and-pepper method. Applying salt-and-pepper noise is a process of changing a fraction of the pixels in the image to their minimum or maximum values (0 or 1) [59]. The vast majority of the image remains unchanged. In contrast, approximately 1% of the pixels (chosen at random) were altered to either 0 or 1. The images in the dataset are multispectral (i.e., containing 12 channels). A pixel was chosen to be altered, the value of all the channels was updated at the same pixel.

### 3.2.3. Identifying Crop Yield Prediction Targets

The proposed models presented in this section have a single output: the predicted crop yield per $1000\,\mathrm{m}^2$. However, the target yield is slightly different between some models, requiring some clarification. As the dataset contains how much a farmer has delivered for each crop type, as well as how large areas have been harvested for each type, we can calculate a target yield for each crop type separately, as shown in Equation (3). Another approach is to use the sum of all crops delivered and the area harvested for each farm, resulting in the total yield per farm as shown in Equation (4). The two methods both allow models to be trained to predict farm-scale crop yield. However, they also produce slightly different distributions, which means that predictions of one type are not directly comparable to the other.

$$y = \frac{\text{Grains delivered (kg)}}{\text{Area harvested }(1000\,\mathrm{m}^2)} \tag{3}$$

Equation (3) illustrates how the ground truth was calculated for each sample for models that take crop type as input.

$$y_{\text{total}} = \frac{\text{Sum of all grains delivered (kg)}}{\text{Total area harvested }(1000\,\mathrm{m}^2)} \tag{4}$$

Equation (4) illustrates how the ground truth is calculated for each sample of the Single Image CNN. The Single Image CNN (see Section 3.4.1) is the only model using the total yield per farm, while all other models are trained on the crop-specific yield targets. As some farmers deliver multiple crop types each year, some samples are duplicated for each calculated crop yield target. The models are given the crop type as input to differentiate between them. By providing the same image or weather input but with different crop type inputs, the aim is to force the models to learn yield characteristics for each crop type. In the next subsections, we explain the baseline approach and newly proposed approaches for this study.

### 3.3. Prediction Model Using Weather Data

As mentioned earlier in this section, weather data directly include information for two of the four main factors of plant growth: precipitation and temperature [14,56]. Training a deep neural network on the weather data allows us to verify the utilisation and relevancy of these data.

#### Baseline Approach: The Weather DNN Model

A deep neural network from the preliminary experiment serves as a baseline for our other models [53]. The model is a feed-forward neural network consisting of an input layer, three densely connected layers with Tanh activation of 512, 128, and 64 units, respectively, and one output layer at the end. After the two hidden layers, there are 10% and 25% dropouts, as that seems to give the best generalisation. The model has 883 input features, 856 (96.9%) of which are weather features. The remainder is historical, positional and other relevant features such as the cultivated area and crop type. The readers can access the detailed explanation of this experiment in [53].

### 3.4. Prediction Models Using Satellite Images

Satellite images are remotely sensed data collected by earth-observing satellites. These images are available globally, cost effective, and include detailed high-resolution observations of the earth. The multispectral satellite images contain detailed crop growth and plant health information, traditionally extracted using handcrafted vegetation indices. By training on per-farm satellite images, models can automatically extract important, relevant features for crop yield. The models discussed in the upcoming subsections use raw satellite images to extract relevant features and make crop yield predictions.

#### 3.4.1. Initial Experiment 1: The Single Image CNN Model

The initial model using satellite images was a simple CNN. This model aims to prove the concept of this study and indicates whether satellite images of farms in Norway contain some information that can be used with deep learning to predict grain yield. To keep the model as simple as possible, it takes *one* multispectral image as input and makes yield predictions based on this. The ground truth is calculated by summing all-grain deliveries and dividing this by the total area harvested for each farm, resulting in a grain yield target specified in kg/1000 m², and can be seen in Equation (4).

The model input layer is by default $100 \times 100 \times 12$, meaning images of $100 \times 100$ pixels and 12 channels deep, see Figure 4. In specific experiments, such as adding mask as a channel and cropping the images, the input layer is adjusted slightly to accommodate images of size $90 \times 90$ or 13 instead of 12 channels. As can be seen in Figure 4, the CNN layers are made out of three pairs of 2D convolutional layers and 2D max-pooling layers. Each pair has an increasing number of $3 \times 3$ convolutional filters with ReLU activation applied to the input (16, 32, and 64, respectively) and a max-pooling of size $2 \times 2$ reduces the output dimensions at each step. Next, the final max-pooling output is flattened before being fed into a dense layer of 32 units with a single dense layer at the end.

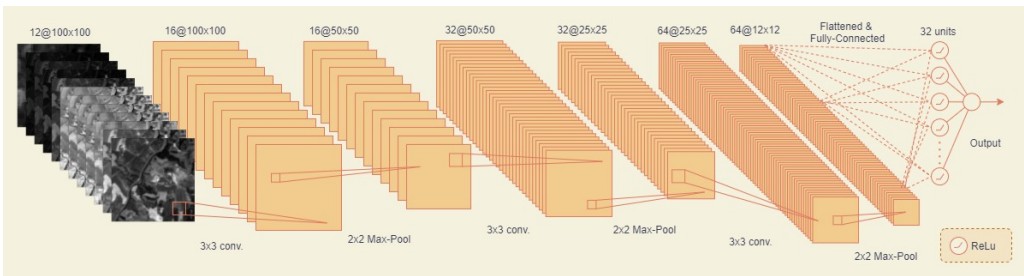

**Figure 4.** Initial experiment 1: single image CNN model architecture.

#### 3.4.2. Initial Experiment 2: A Multi-Temporal CNN–RNN Model

As examined by Jiang et al., the developments of crop phenology play an essential role in the eventually harvested yield [17]. Grain crops are typically planted either during autumn or in the spring, and the growth progress from seedlings to mature harvestable crops is not constant or fixed through time [19]. By training a model on multi-temporal images from the growing seasons, the aim was to achieve higher accuracy by analysing the changes that occur over time. The proposed model is a convolutional and recurrent neural network that takes an image time-series as input and outputs the predicted yield/1000 m². The model uses a similar CNN architecture to the single image CNN model described earlier for each image. The CNN output is then fed into a GRU encoder network together with a one-hot encoding of the crop type and a normalised field area, which is duplicated for each time step.

The architecture of the CNN used for each time step has the same convolutional and max-pooling layers as the single image CNN, see Figure 5. A single 64 unit fully connected output layer replaces the two dense layers, effectively reducing each image to a 64 element vector. The same CNN weights are reused for each time step, meaning that the network size is independent of sequence length. The CNN output is concatenated with crop type

and area and fed into a GRU encoder with 128 units. The GRU encoder output is fed through a single fully connected layer with ReLU activation, and the outcome is a single neuron with no activation function (linear).

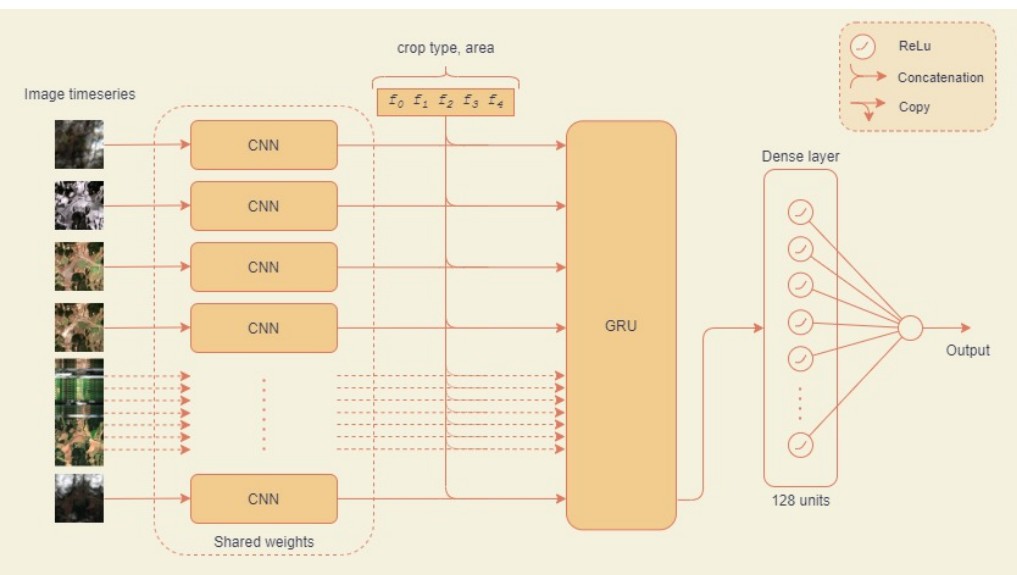

**Figure 5.** Initial experiment 2: multi-temporal CNN–RNN model architecture.

### 3.5. Prediction Models Using Combined Satellite Images and Weather Data

Between the Weather DNN and multi-temporal CNN–RNN, all the previously described data are used as feature inputs. However, the models train and predict individually, so the models cannot learn any patterns that only appear when both satellite and weather data are combined. For this reason, one LSTM and two hybrid models were created and explained.

### 3.5.1. Initial Experiment: LSTM Model Using Handcrafted Features

Following the work of Johnson and Bendetti [14,15], the model evaluates the use of vegetation indices and weather data to predict yield on a farm scale. As mentioned earlier in this section, we utilised and synchronise 7-day interval Sentinel-2 time series data with weather features, i.e., temperature and precipitation, such that the time window matches with the Sentinel-2 data. We calculated min, max, and mean values for temperature features, resulting in three temperature features per interval. The precipitation is summed for each group, resulting in one precipitation feature for the total precipitation per interval. Image masks are applied to each image so that only cultivated crops remain. We also calculated the vegetation indices as described earlier in this paper [14,15]. Each vegetation index is calculated for all remaining pixels, and the mean value represents the actual vegetation index. The specific vegetation indices include NDVI, WDRVI, NDWI, and NDMI (See Table 3 for details), resulting in four vegetation indices per interval.

We implemented a basic LSTM based architecture using handcrafted features, see Figure 6. The LSTM encodes the time-series-based data across the growing season. Next, the encoded growing season data are concatenated with the additional farm-related properties in dense layers. The final output is the predicted yield in $kg/1000\,m^2$. The LSTM consists of 30 cells (time steps), where each of these cells contains 32 units. After the LSTM layer, there is a dropout layer of 25%, a dense layer of 32 units, and another dropout layer of 10%. Next, the farm-related property input layers are concatenated into the network before a 32-unit dense layer and a final single neuron output. The LSTM units use the activation function Tanh, and all dense layers use ReLU.

**Table 3.** Handcrafted features used as input. The bands (B) are Sentinel-2 specific.

| Name | Abbr. | Formula | Description |
|---|---|---|---|
| Normalised Difference Vegetation Index | NDVI | $\dfrac{B8 - B4}{B8 + B4}$ | Indicator of green leaf area, giving a measurement of healthy green vegetation in any given pixel. First used by Rouse et al. [11] |
| Wide Dynamic Range Vegetation Index | WDRVI | $\dfrac{\alpha * B8 - B4}{\alpha * B8 + B4}$ | Modification of the NDVI with an extra weighting coefficient parameter. Increased sensitivity when areas with moderate to high biomass are investigated [60]. |
| Normalised Difference Water Index | NDWI | $\dfrac{B3 - B8}{B3 + B8}$ | A measurement that is sensitive to changes in water content of vegetation [61]. |
| Normalised Difference Moisture Index | NDMI | $\dfrac{B8 - B11}{B8 + B11}$ | Indicator of the water content of vegetation. Effectively similar to that of NDWI, but calculated using other aspects of the spectrum [61]. |

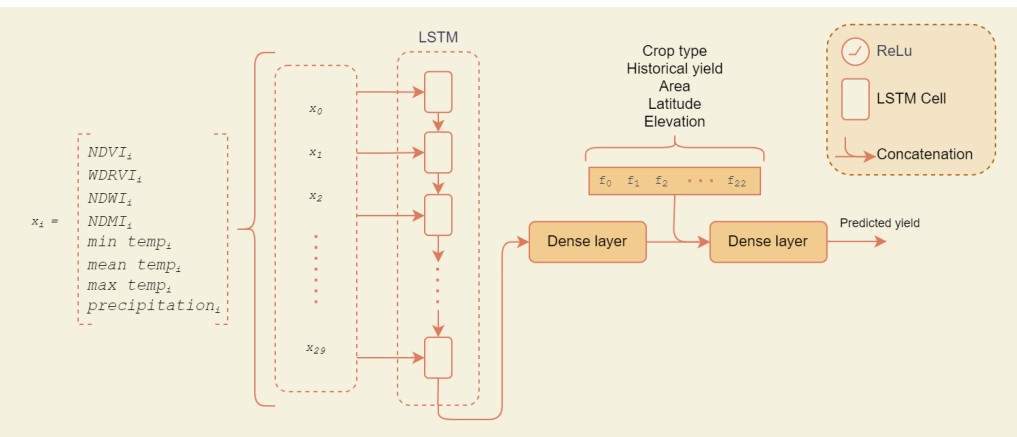

**Figure 6.** Initial experiment: architecture of LSTM model Using handcrafted features.

### 3.5.2. Novel Approach 1: Pre-Trained Hybrid Model

The pre-trained hybrid model (see Figure 7) combines the weather DNN and multi-temporal CNN–RNN by concatenating the outputs of the second to last layers and feeding it into a deep neural network consisting of three fully connected layers. The first two layers of the combined network use ReLU activation and have 64 neurons each, followed by 10% dropouts, and the last layer of the combined network is a single neuron that outputs the predicted crop yield. The weather DNN and multi-temporal CNN–RNN are trained separately, first the multi-temporal CNN–RNN, followed by the Weather DNN. Layers that include the second to last layers of the pre-trained models are locked so that their weights remain unchanged. Finally, the complete hybrid model is trained, combining the value of both sub-networks.

The dataset was prepared so that the training/validation split would remain constant through all stages to avoid leaking validation data into training data between the three stages. Because the weather data have significantly more samples that increase error when excluded, custom separate dataset generators provide each stage with all relevant features and samples while keeping the training/validation split the same for all three.

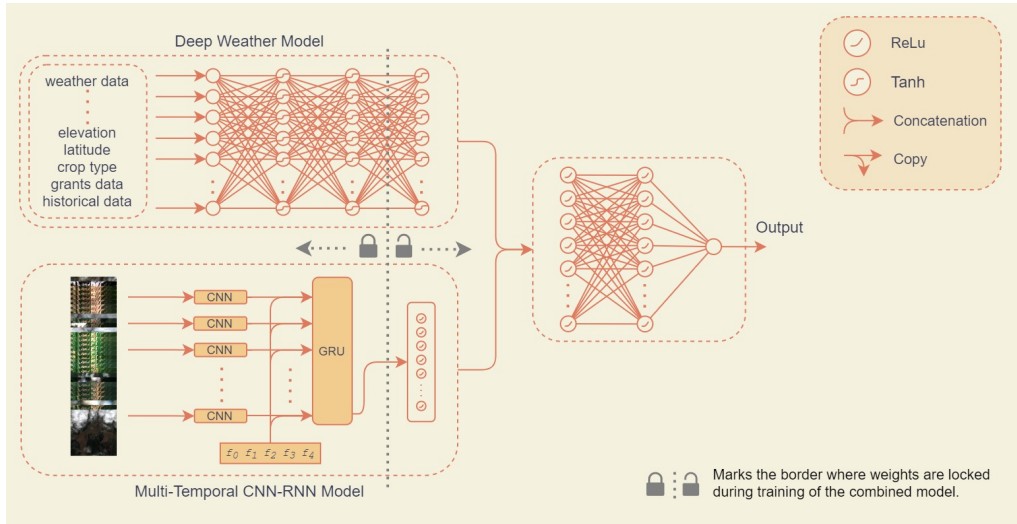

**Figure 7.** Novel approach 1: architecture of Pre-Trained Hybrid model.

### 3.5.3. Novel Approach 2: Hybrid CNN Model

Although the pre-trained hybrid model combines all available features and data for each farm to make predictions, the method makes the model slightly cumbersome to train. The architecture is inefficient because some of the learning in the two individual models is discarded when the last layers are skipped in the combined model. The second proposed hybrid model, shown in Figure 8, is trained in a single stage. The model has fewer trainable weights than the first and combines satellite images and weather data in 7-day time steps. That allows the model to process a single sequence using both data sources. The model shares much of the architecture with the multi-temporal CNN–RNN model and incorporates weather data through a one-dimensional convolution, allowing both satellite images and weather data to be encoded in 7-day time steps. The model then combines the output from the encoded satellite images and the 7-day weather data by concatenating both vectors for each time step. The concatenated vectors are then fed into a GRU encoder, which encodes the entire sequence into a 128 length vector. A fully connected layer with ReLU activation and a single output neuron predicts the crop yield per 1000 m$^2$.

Both weather and satellite data are captured from 1 March to 1 October for each sample. However, satellite images have a temporal resolution of 7 days, while the weather data have a higher temporal resolution of 1 day. To combine these two data sources, we apply a one-dimensional convolutional layer, with a size and stride of 7, on the weather inputs. The size and stride used in the one-dimensional convolution mean that the weather time series is reduced to 30 vectors, effectively encoded as 7-day intervals. The one-dimensional convolutional layer has 64 filters. Seven days of weather data are encoded as a vector of length 64, the same size and temporal resolution as the CNN outputs from the satellite images sequence.

For the trained models with satellite image time series, the size of each training sample (without cropping) is at least $100 \times 100 \times 12 \times 30 = 3,600,000$ parameters when not counting additional inputs to the models. We suspect a large number of parameters for each training sample is the reason the models are prone to overfitting [62]. A minimum of 5 h of training time per epoch with the augmented dataset would be expected using the available hardware. During this time, the model might have already started to overfit the training samples. It is impossible to know exactly when the model performed best without testing on validation samples more frequently. Validation is typically done at the end of each epoch. Because data augmentations increase the dataset size and, thus, the number of samples in an epoch, the time between validations also increases. We solve this by artificially reducing the size of each epoch by taking a random subset of the samples instead of the whole set. This leads to more frequent validation runs, which allows us to better monitor how well the model is learning. Figure 9 illustrates how stochastic epoch

sampling provides higher resolution on the monitored loss values. In all our experiments, training is stopped when validation loss has not improved for a set number of epochs, and the model weights are restored from the epoch with the best validation loss. The stochastic epoch sampling ensures we can restore the model weights from the best point in time while also preventing models from training unnecessarily long.

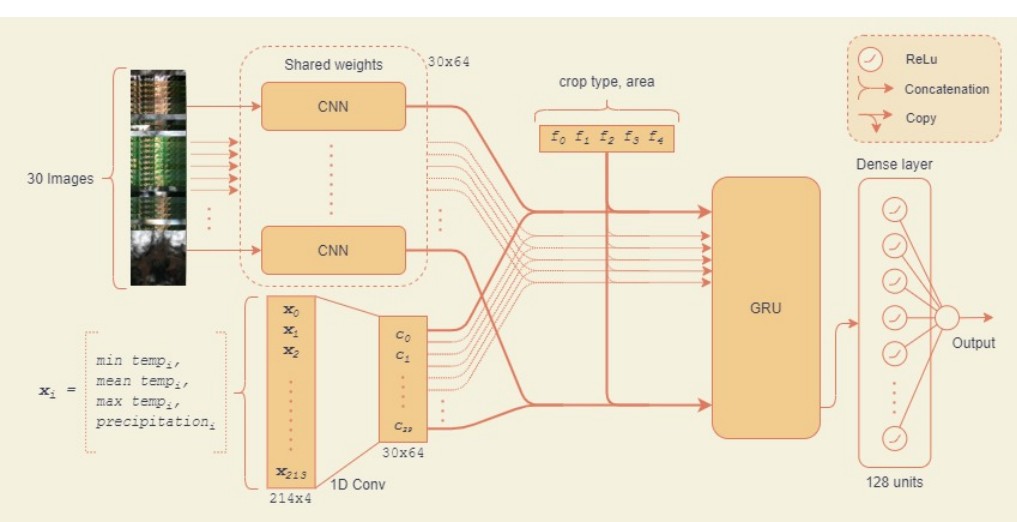

**Figure 8.** Novel approach 2: architecture of Hybrid CNN model.

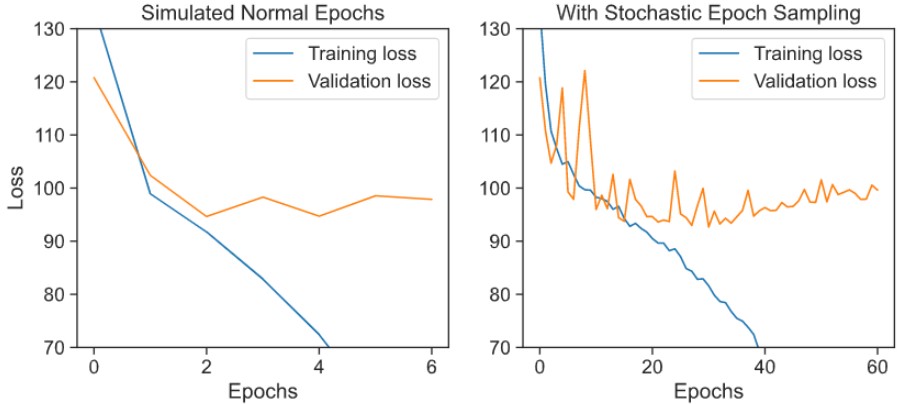

**Figure 9.** Illustration of stochastic epoch sampling.

## 4. Results

This section includes preliminary experiments conducted to evaluate the use of multispectral imagery to predict yield, give indications of the effectiveness of data augmentation methods mentioned in the previous section, and test the use of image masking. In addition, we show the performance and comparisons of baseline and novel approaches discussed in the earlier section. We scale the model up to predict on a per commune basis so that a comparison can be made between the proposed approaches and the work of Sharma et al. [8], discussed in Section 1.2.3 earlier in this paper.

### 4.1. Preliminary Experiments on Multispectral Images

The experiments use the single image CNN, which is built to be a simple CNN-based model so that the impact of the different experiments becomes clear. As explained in Section 3.4.1, this model did not differentiate between crop types and was trained to predict total crop yield per 1000 m$^2$ for each farm. This means the Mean Absolute Error (MAE) cannot be directly compared to that of the other models in the study.

### 4.1.1. Evaluating the Use of Multispectral Imagery for Crop Yield Prediction

The first experiment conducted with the multispectral images checks whether a CNN can predict the yield with better accuracy than if one just predicted the average. By calculating the $y_{total}$ (all grains delivered in kg divided by total area harvested) per farm, the mean yield per $1000\,m^2$ across the training dataset results in $381.2\,kg/1000\,m^2$. When using the mean yield per $1000\,m^2$ as the prediction on the validation data, the MAE is $134.3\,kg/1000\,m^2$. Anything lower than this means that the model has found some relevant features from the multispectral images. The single image CNN is trained using one image per year for each farm, which results in a validation loss of $96.4\,kg/1000\,m^2$ MAE. This indicates that the model can learn relevant features from the multispectral images, validating the use and further exploring satellite images.

### 4.1.2. Optimal Week to Predict Crop Yield

The continued development and stages of crop growth play an essential role in the harvested yield, which could mean that selecting different weeks as input for the single image CNN will increase or decrease the performance. The model is trained separately for each chosen period of a week to determine the most relevant information for each year, i.e., 1 March to 1 October. There are measurable variations in the performance across the growing season. The period from week 26 to 29 seems to be when the model performs the best, which corresponds to roughly 25 June to 22 July. Interestingly, this finding is similar to what Basnyat et al. found to be the optimal time to use remote sensing for crop grain yield on the Canadian prairies, which was between 10 and 30 July [63]. Based on these findings, week 26 is chosen for all single-image based experiments going forward.

### 4.1.3. Effects of Data Augmentation

The model described earlier showed signs of overfitting on the initial runs; hence, we added data augmentation to reduce overfitting and increase the overall performance. The data augmentation methods used are cropping, rotating, and adding noise (more details are in Section 3.2.2). The datasets are split into training and validation sets before augmenting the training data. When cropping, the validation images also have to be cropped to match the model requirements of $90 \times 90$ images, and the centred $90 \times 90$ crop is applied. For these specific experiments, the rotations are 90°, 180°, and 270°. Each of the augmentation methods is tested separately. There are improvements both regarding less overfitting, and lower loss overall. By evaluating the best-achieved loss, as shown in Table 4, we see that the salt-and-pepper brings a modest improvement of 3.3%, while the improvements from cropping and rotating are both at about 7.1%. In addition, when combining rotating, cropping and salt-and-pepper, the validation loss further improves to 10.1% overall. These positive results of augmenting the satellite images suggest the models would improve as more years of data become available.

**Table 4.** Effects of the augmentations. The validation loss is the mean of three separate runs.

| Augmentation | Validation Loss (Mean) | Improvement over Original (Percent) |
|---|---|---|
| Original (No augmentation) | 89.8 | |
| Salt-and-pepper noise | 86.8 | 3.34% |
| Rotating | 83.4 | 7.13% |
| Cropping | 83.3 | 7.24% |

### 4.1.4. Effects of Image Masking

By quantifying the effects of image masking, we can answer the second hypothesis, i.e., whether accurate field boundaries can significantly increase accuracy. In this context, field boundaries and satellite images can be applied using pixel masks. In theory, the masks should remove or highlight the cultivated crops and enable the models to focus on crop-specific features. This experiment tests the use of image masks and the two proposed

methods of applying these, as explained in detail in Section 3.1.5. Figure 10 shows the results of both masking techniques compares to the original with no mask. Both applied and, as a channel, seems to aid the model and improves the validation loss achieved. Surprisingly, masks as a channel perform the best by quite a margin. This could be because adding masks as a channel primarily adds information to the image in a separate channel, suggesting that the environment around and close to the cultivated fields is also relevant.

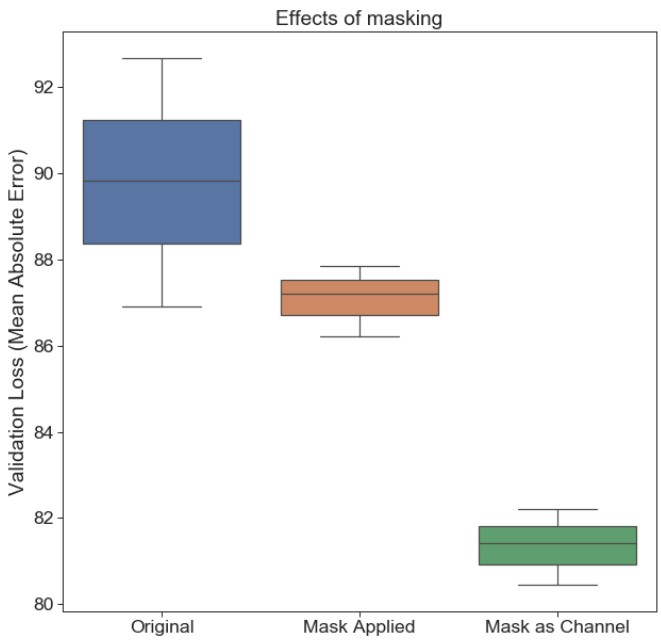

**Figure 10.** The effects of masking techniques on MAE.

*4.2. Comparison of Crop Yield Prediction Models*

We evaluate the proposed models by comparing the achieved mean absolute error in Table 5, which shows that the two best performing models, i.e., the two hybrid models, incorporate both weather data and satellite images. The single image CNN model is left out of this comparison, as the prediction targets for this model were total yield per $1000 \, \text{m}^2$. All other models predict crop yield per $1000 \, \text{m}^2$ for each crop type individually, which belongs to different distributions and is difficult to compare directly.

**Table 5.** Best mean absolute error achieved for each model.

| Model | Mean Absolute Error (kg/1000 m$^2$) |
|---|---|
| Weather DNN | 83.04 |
| Multi-temporal CNN–RNN | 80.52 |
| Handcrafted features in LSTM | 82.29 |
| Hybrid 1: pre-trained hybrid | 77.53 |
| Hybrid 2: hybrid CNN | 76.27 |

4.2.1. Baseline Approach: The Weather DNN Model

The baseline model, the weather DNN model, achieves a mean absolute error of $83.04 \, \text{kg}/1000 \, \text{m}^2$ (see Table 5) using daily interpolated temperature and precipitation values as described in Section 3.1.3. The interpolated weather data results in an improvement of around $10 \, \text{kg}/1000 \, \text{m}^2$ compared to previous results using only measurements from the nearest weather station [53]. The weather DNN results represent the benchmark for which we test our other models on the first and third hypotheses: whether satellite images can predict yield accurately and if satellite images combined with weather data increase the accuracy further. The multi-temporal CNN–RNN validates the first hypothesis, and the

two hybrid models show that combining satellite images and weather data are also beneficial. The second hypothesis, concerning accurate field boundaries, are tested by running the best performing model, the hybrid CNN, both with and without masks, showing that masking of fields improves accuracy (see Table 6). Further results and insights from the other models are presented and discussed in the following subsections.

**Table 6.** Hybrid CNN results with and without masks.

| Mask Type | Mean Absolute Error (kg/1000 m$^2$) |
|---|---|
| No mask | 86.69 |
| Mask as channel | 76.57 |
| Mask applied | 76.27 |

### 4.2.2. Initial Experiment 1: Multi-Temporal CNN–RNN

The multi-temporal CNN–RNN model utilises the satellite image dataset to the fullest by processing all the 30 images for each sample to yield a prediction. The model achieves a mean absolute error of just above 80 kg/1000 m$^2$ (see Table 5), an improvement from the Weather DNN. However, compared with the weather DNN, the multi-temporal CNN–RNN is not given a farm's previous grain deliveries or positional data; the model is trained using only multispectral satellite images and the crop type and area encoding. Figure 11 shows the training and validation loss for the multi-temporal CNN–RNN model. The training was performed using images with a pixel mask added as a separate channel, given the best results in the single image CNN experiments.

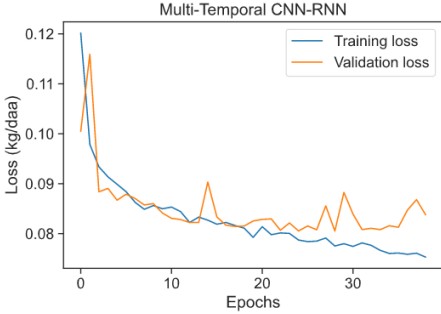

**Figure 11.** Training and validation loss for the multi-temporal CNN–RNN model.

While a multispectral satellite image contains more information than a temperature and precipitation measurement, a significant drawback is cloudy images, which can sometimes constitute a large portion of the images in the 30 image time-series samples. The model still predicts the crop yield more accurately than the weather DNN, suggesting that it can successfully extract valuable information from the good quality images while ignoring the noise generated by cloudy images. This is perhaps due to the GRU-encoders ability to control how much each input should contribute to the encoding at each time step with the update gate (GRU explained in Section 2).

### 4.2.3. Initial Experiment 2: Handcrafted Features in LSTM

By condensing the relevant sequential time series data into sequences of vectors suited for an LSTM, the LSTM trained on handcrafted features achieves a mean absolute error of 82.29 kg/1000 m$^2$. The model is tested with three sets of sequential inputs: weather, vegetation indices, and a combination of weather and vegetation indices. Surprisingly, weather alone only results in an MAE of 93.63 kg/1000 m$^2$, while vegetation indices see an improved MAE of 83.01 kg/1000 m$^2$. With both weather and vegetation indices, the MAE improves to 82.29 kg/1000 m$^2$. See Figure 12 for a comparison of the training.

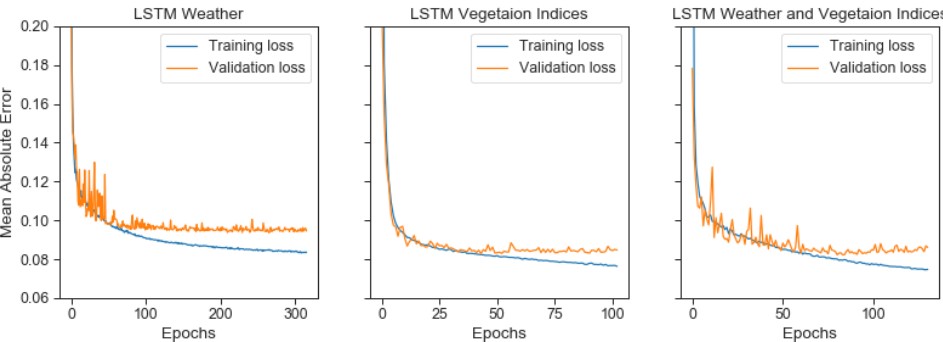

**Figure 12.** Training and validation loss for the LSTM model.

The relatively poor MAE results of weather features alone can indicate that weather features condensed into four measurements per week remove necessary granularity. Overall, the MAE achieved with this model is a modest improvement over the Weather DNNs results. One contributing factor to this result can be that although we have information on which fields the farmers cultivate each year, we cannot differentiate the type of crops in the fields. By not making vegetation indices for just the relevant types of crops, they may be affected by fields of grass, potatoes, and vegetables.

### 4.3. Novel Approach 1: Pre-Trained Hybrid Model

The LSTM model with handcrafted features shows that combining vegetation indices derived from satellite data with weather data only improves predictions marginally. On the contrary, the pre-trained hybrid model shows that combining the full weather DNN and the multi-temporal CNN–RNN, a definite improvement is achieved compared to each model's results individually. Both the Weather DNN and multi-temporal CNN–RNN are trained individually before training the hybrid, which reduces the number of epochs needed to train the hybrid, shown in Figure 13. The lowest achieved loss is 77.53 kg/1000 m$^2$, 6.6% lower than the Weather DNN and 5.3% lower than the multi-temporal CNN–RNNs individual results.

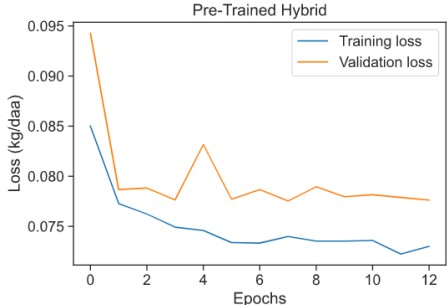

**Figure 13.** Training and validation loss for the pre-trained hybrid model.

The pre-training of the two combined models may explain the shorter training time for the pre-trained hybrid. However, it may also indicate that the hybrid model cannot find any valuable complex patterns between weather and satellite data, perhaps due to its architecture.

### 4.4. Novel Approach 2: hybrid CNN Model

The hybrid CNN is the best performing model. It manages so with fewer parameters and features than the pre-trained hybrid, which contains additional features, such as a farmer's previous deliveries (historical yield). We attribute the hybrid CNNs improvement to its more natural architecture that combines weather and image data into a single sequence encoding. Compared to the multi-temporal CNN–RNN, which only looks at satellite

images, the addition of weather data may allow the model to estimate plant growth where cloudy images would usually blind the model.

As the best performing model, we also test the model both with and without masking. As with the single image CNN experiment, we find that masking of the image provides significant improvements compared with no mask (see Table 6 and Figure 14). We identified that accurate field boundaries are beneficial for accurate crop yield predictions on a farm scale. However, in contrast to the single image experiment with masking results, the hybrid CNN model performs almost equally with both masks as a channel and masks applied. Although we have no conclusive answer why, one reason may be that the model starts overfitting earlier than the Single Image CNN and thus never reaches its full potential.

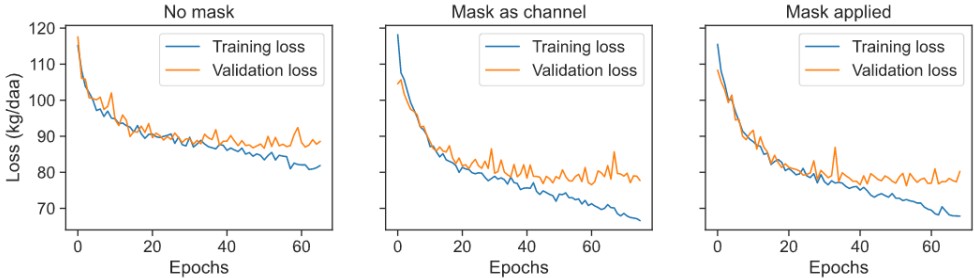

**Figure 14.** Training and validation loss for the hybrid CNN.

As an additional analysis of the model's accuracy, we present a quantile–quantile plot (Q–Q plot) in Figure 15 which compares the prediction output distribution with the actual distribution. The Q–Q plot shows the model learns an excellent approximation of the actual distribution of crop yields from the validation set. However, very low and very high yields seem to be more challenging to predict.

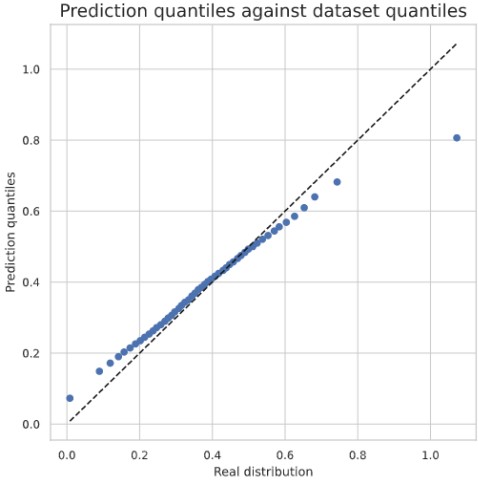

**Figure 15.** Hybrid CNN prediction quantiles versus real quantiles.

To visualise and better understand the model's predictions, we discretise predictions by grouping them into bins, then plotted as a heat map showing the cross-tabulation between prediction output and actual yield values. Figure 16 shows the discretised prediction outputs in equal-width bins, which shows that the model predicts well for the most common values. The figure may also explain why the model has difficulty predicting very high and very low yields due to fewer samples to train on the model. A more balanced view of the prediction outputs is shown in Figure 17 where predictions are grouped into bins created from percentiles instead of a fixed width. The percentile bins show that predictions are centred roughly around the actual values on the diagonal, even for values in the top and bottom 10%.

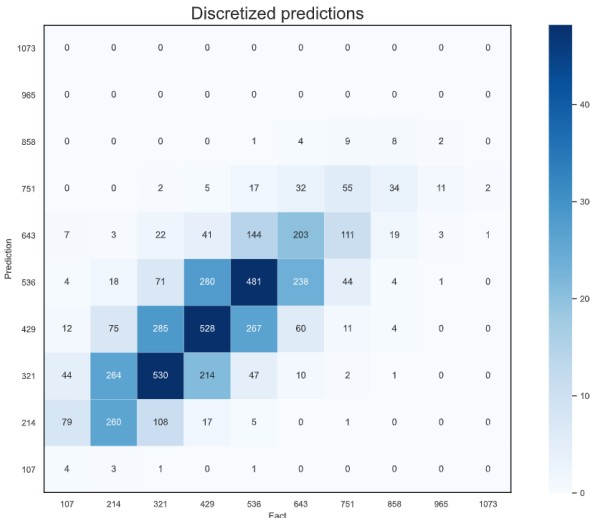

**Figure 16.** Discretised prediction output from the hybrid CNN. Predictions are binned into ten bins. The axes' labels mark the upper bound of each bin.

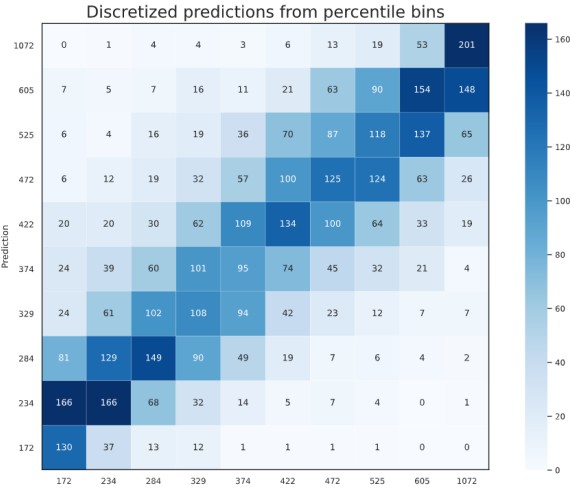

**Figure 17.** Discretised prediction output from the hybrid CNN in percentile bins. Predictions are binned into ten percentile bins (10, 20, ..., 100) derived from the actual distribution. The axes' labels mark the upper bound of each percentile bin.

*4.5. Early Predictions*

In Hypothesis 4, we propose that satellite and weather time series data can be used to predict crop yields before harvest. To test how much the early and middle periods of the growing season effect crop yield, we reuse the best performing model, the hybrid CNN with masks applied, to predict crop yields without the whole data time series. The single image experiments have already shown that satellite images from week 26 appear to contain the most relevant information out of all the weeks, implying that data after week 26 might not be as essential to crop yield. We can compare early predictions versus predictions on the whole time series by training the hybrid CNN on shorter time series. Table 7 show the mean absolute errors achieved when limiting the amount of data to 12 and 17 weeks of data, roughly equal to mid-May and late-June predictions, respectively.

**Table 7.** The mean absolute error achieved at different times with early predictions. The error increases by 7.66% for late-June predictions and 20.89% for mid-May predictions, compared to predictions made using the whole season.

| Input | Description | Mean Absolute Error | Change |
|---|---|---|---|
| Weeks 10–39 | Full season | 76.27 kg/1000 m$^2$ | – |
| Weeks 10–26 | Late-June | 82.11 kg/1000 m$^2$ | +7.66% |
| Weeks 10–21 | Mid-May | 92.20 kg/1000 m$^2$ | +20.89% |

As expected, the error increases when predicting earlier in the growing season. The late-June predictions, which include all weeks up to and including week 26, have a moderate increase in error that is still lower than predictions made by other models in the full season. Figures 18 and 19 show the predictions in percentile bins for late-June and mid-May, respectively. Mid-May predictions show a clear reduction in accuracy compared to both late-June and full-season predictions (see Figure 17), as the model struggles to differentiate between low and medium yields.

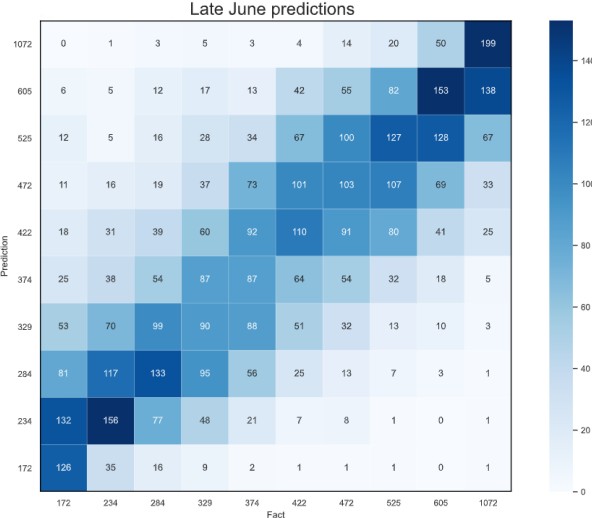

**Figure 18.** Early Predictions: Late-June.

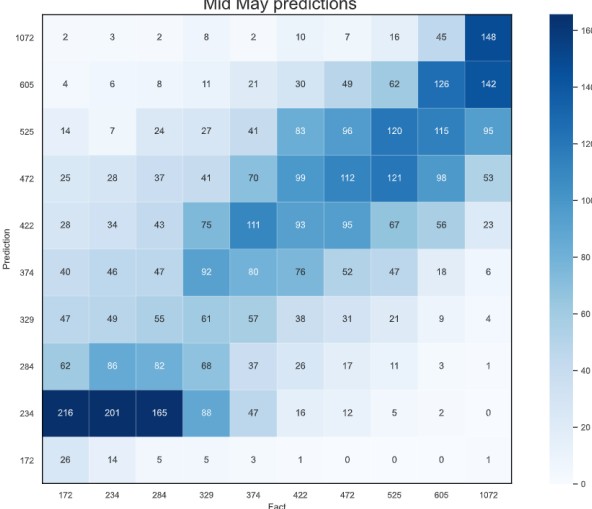

**Figure 19.** Early Predictions: Mid-May.

### 4.6. Predictions as Regional Analytics

To the best of our knowledge, no prior work predicts real-world farm-scale crop yields on a large scale, making it difficult to compare our results with the current state-of-the-art [22] crop yield predictions. Sharma et al. [8] predict wheat yield at a tehsil scale, a small administrative unit in India and achieve an RMSE of between 4.8 and 33 kg/1000 m$^2$ when trained in different states in India. To compare our results, we aggregate predictions made on a farm scale up to the lowest administrative unit, the Norwegian commune, by assuming a mean farm-scale prediction per year for each commune. As some communes have very few samples, we take the 100 communes with the highest number of samples in our dataset to analyse (the number of samples in the top 100 communes ranges from 32 to 152, with a mean of 60). With this approach, our best performing model achieves a nationwide commune-scale crop yield prediction with an MAE of 23.35 kg/1000 m$^2$ and an RMSE of 30.81 kg/1000 m$^2$. This indicates that its accuracy is in the range of the predictions made by Sharma et al. on a tehsil scale in India (as there are too many uncontrolled variables between these projects, this comparison only provides a rough idea of the performance level of two models that were not meant to predict the same thing). Figure 20 shows the relationship between predicted and actual commune-scale crop yield, made by aggregating farm-scale yields.

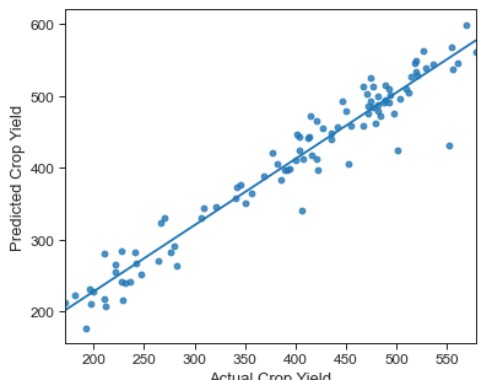

**Figure 20.** Relationship between actual and predicted crop yields on a commune-scale.

### 5. Discussion

Evaluating the initial experiments, the single image CNN can find a correlation between the multispectral satellite images and the yield of the farms. Further, the data augmentation methods of rotating, cropping, and adding noise to the images seem adequate, indicating that the model's accuracy could improve given additional data. The application of image masks also provided positive results, showing that highlighting or keeping only the farms' cultivated fields in the images enables the model to focus on the relevant portions of the image.

The best performing model is the hybrid CNN, which utilises both weather data and raw multispectral satellite images as its input and improves 8% over the baseline Weather DNN. Seemingly by the findings of You et al. [9] and Sharma et al. [8], using raw multispectral images, outperforms the model using handcrafted vegetation indices. The hybrid CNN can also make early, in-season predictions, though the error increases when data decreases. To make the results of the proposed hybrid CNN comparable to earlier works, we average the model's per-farm predictions to predict on a per-commune basis. We see that the per-commune predictions of the hybrid CNN are comparable to the latest state-of-the-art in crop yield predictions.

### 6. Conclusions and Future Work

In conclusion, this study explores using satellite data for crop yield prediction, reviewing traditional and new methods of extracting relevant information from satellite imagery.

A new dataset with real-world per-farm samples, created by combining many data sources, enables deep learning for farm-scale crop yield prediction. Multiple models are proposed to test different hypotheses and optimise prediction accuracy. The most accurate model is a deep convolutional, recurrent, and hybrid model. It combines multispectral satellite images and weather data to predict crop yields. The model is a first of its kind in predicting farm-scale crop yields to the best of our knowledge. In addition, the aggregated farm-scale predictions to a commune scale are presented, and the proposed model achieved comparable results to the current state-of-the-art crop yield predictions. The hypotheses defined for this research study are tested and explained, along with a brief conclusion for each, as follows.

**Hypothesis 1.** *Satellite images of farms and their surroundings can be used to accurately predict farm-scale crop yields.*

Satellite images were used to predict farm-scale crop yields, and the results show that models using satellite images improves the prediction accuracy over the baseline model using weather data.

**Hypothesis 2.** *Accurate field boundaries along with satellite images increase crop yield accuracy significantly.*

We show that accurate field boundaries, represented using pixel masks and satellite images, significantly improve prediction accuracy using the best performing model. The best performing model saw a 14% reduction of mean absolute error with the pixel masks, from $86.69 \, \text{kg}/1000 \, \text{m}^2$ without to $76.27 \, \text{kg}/1000 \, \text{m}^2$ with a mask.

**Hypothesis 3.** *Prediction accuracy can be further increased by combining satellite images and meteorological data.*

Prediction accuracy was consistently better with models that incorporate both weather data and satellite images, suggesting that both contain some information that the other does not.

**Hypothesis 4.** *It is possible to predict farm-scale crop yield earlier in the growing season with some reduced accuracy.*

By training the best hybrid model using data from earlier in the season, we show that late-June predictions can be made with a moderate increase in mean error (+7.66%). In contrast, mid-May predictions are significantly less accurate with an almost 21% error increase.

Given that this is a novel application of neural networks in a domain where data are limited and noisy, many untested methods and data sources could improve prediction accuracy or achieve similar results more efficiently. For future work, we suggest further exploration of the following:

- **Improving generalisation:** while our models show that accurate farm-scale crop yield predictions are possible with deep learning, the majority of models start to exhibit overfitting when training, even with the data augmenting methods used. This suggests that even higher accuracy might be possible given more data or using other known methods for reducing overfitting and increasing generalisation, such as batch normalisation.
- **Remote sensed temperature:** land surface temperatures derived from satellite sensors have successfully been used in US county-level predictions [14]. They provide temperature values that should be closer to the actual temperature at the farm compared to interpolations between sensors that are typically many kilometres away from the farm. Such values could replace temperature interpolation or be used to improve weather interpolation further.

- **NIBIO field data:** apply field data gathered by NIBIO to the models. Adding soil quality and water storage capacities could add meaningful information about crop yield potential in individual fields.
- **Additional sources for satellite images:** this study used Sentinel-2 as the source for the satellite images. It could be positive to introduce additional sources for satellite images to complement the Sentinel-2 dataset. Additional sources for satellite images could increase the frequency of the satellite images throughout the growing season and give higher resolution multispectral images. Some specific satellites of interest could be Landsat 8, WorldView-3, and PlantScope.

**Author Contributions:** Conceptualisation, M.E., E.S., B.L.O.S., S.A. and M.G.; methodology, M.E., E.S. and B.L.O.S.; software, M.E., E.S and B.L.O.S.; validation, R.G., S.A. and M.G.; formal analysis, M.E., B.L.O.S. and E.S.; investigation, M.E., E.S., B.L.O.S. and S.A.; resources, M.E., E.S. and B.L.O.S.; data curation, S.A.; writing—original draft preparation, M.E., E.S. and B.L.O.S.; writing—review and editing, R.G. and M.G.; visualisation, M.E., E.S., B.L.O.S., S.A., R.G. and M.G.; supervision, M.G.; project administration, S.A., R.G. and M.G.; funding acquisition, M.G. All authors have read and agreed to the published version of the manuscript.

**Funding:** The work has been carried out as part of the Norwegian research council-funded project 309876 KORNMO—production optimisation, quality management, and sustainability through the grain value chain.

**Institutional Review Board Statement:** Not applicable.

**Informed Consent Statement:** Not applicable.

**Data Availability Statement:** The data utilised in this study were collected from public resources, and the required links are provided in Section 3.1 in the manuscript.

**Acknowledgments:** We would like to thank the Faculty of Engineering and Science and the CAIR Research lab at the University of Agder, Norway, for allowing us to conduct the research on this topic. We also thank Felleskjøpet Agri SA, who coordinated the KORNMO project—production optimisation, quality management, and sustainability across the grain value chain.

**Conflicts of Interest:** The authors declare no conflict of interest.

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
