# Peer review of "Farm-Scale Crop Yield Prediction from Multi-Temporal Data Using Deep Hybrid Neural Networks"

_agronomy, doi:10.3390/agronomy11122576_

Round 1

Reviewer 1 Report

The article is very interesting and contains valuable research material. The manuscript has been meticulously prepared in terms of editing, but is not free of factual and methodological shortcomings. It needs revision.

The Introduction chapter requires major additions regarding the subject of the study.

The Methodology chapter contains elements of the literature review. It is an incorrect structure.

The chapter Results needs to be reorganized.  Some sentences are not descriptions of results, but statements and conclusions.

The Discussion chapter is very modest. Two literature items are referred to, including one already cited.

The Conclusion chapter is a mixture of discussion and conclusion.

General and specific comments and suggestions:

Page 1:

„Norway, sustainable crop yield production depends on the agro-climatic condi[1]tions, the persistence of rainfall, soil quality, and other infrastructural development”

Please add reference to this sentence

“As the global population has been increasing, it is a significant challenge for the farmers to produce increased quantities and better quality grains.”

Please add reference to this sentence

Page 2

I propose radically shortening section 1.1 or removing it. Information about the project can be found in the article, but it is too detailed.

„We identified that the weather data is helpful to predict farm-scale yield production, and the satellite data is beneficial to predict regional-scale yield production.”

Please add reference to this sentence

Chapter 1.2.

The hypotheses presented for objects other than in Norway have already been studied. The state of knowledge in this area would need to be added.

Chapters 2 and 3 are part review not methodology. The authors themselves write about this:  „In Section 2, we report the history of Norwegian agriculture and grain production. In addition, we report the relevant studies of using remote sensing data and building deep learning models in agriculture improvements. In section 3, we define the current state-of-the-art approaches for the crop yield predictions using remotely sensed data”

I suggest to move it to the Introduction chapter.

Page 4:

“Plant growth and wellness are affected by elements from its surroundings. Accord[1]ing to Oregon State University, the four main environmental factors affecting plants growth are: light, temperature, water, and nutrition”

Please provide references for this sentence and subsequent sentences, but references for peer-reviewed material.

Chapters 2.3, 2.3.1, 2.3.2 contain very basic knowledge.  It proposes to shorten them.

I propose to remove the elaborate structure of sections 4.1; 4.2; 4.4 in which points are used.

Page 11:

“The weather is one of the main external factors that are crucial for farming. Grain farmers depend on periods with little precipitation in the spring so that the fields are dry enough to support heavy equipment for ploughing, harrowing, and sowing. After sowing, the temperature must be stable so that the seedlings sprout and precipitation throughout the summer is required to water the plants”

Please add references to those sentences

Page 12:

“The use of satellite images is the current state-of-the-art for farm-scale crop yield prediction without intrusive and labour-intensive monitoring. Therefore, it is also a significant focus of this research study”

Please add references to those sentences

Page 13:

“Image masking makes it possible to focus on portions of an image that are of interest. Masks can be applied so that it either highlights certain parts or remove irrelevant parts from the image. In the context of this research study, provided we know where the cultivated fields are located, masks can be used on the satellite images to remove everything not registered as a cultivated field or as extra information to highlight where the fields are”

Please add reference to this sentences

“In conclusion, we acquire” – Avoid "We", "I" statements. Use the passive voice.

„In this section, we present the research methodology we have followed for this research study”

Unnecessary sentence. After all, that is what the chapter is called. I suggest deleting it.

Page 14

“Because the memory requirements of the complete dataset are too large to fit in GPU memory or even RAM, images are continuously read from storage”

Why hasn't the computing power of cloud services been used?

General:

1) The use of the unit of area is incorrect. Decares are only used in one place in the world, Norway. I understand the authors' preference, but it is necessary to change it to an area unit according to an international standard, the SI system ( m2 , hm2 or ha).
https://www.bipm.org/en/publications/si-brochure

Author Response

Reviewer 1

Comments and Suggestions from Reviewer

Recommendation: The article is very interesting and contains valuable research material. The manuscript has been meticulously prepared in terms of editing, but is not free of factual and methodological shortcomings. It needs revision. The Introduction chapter requires major additions regarding the subject of the study. The Methodology chapter contains elements of the literature review. It is an incorrect structure. The chapter Results needs to be reorganized. Some sentences are not descriptions of results, but statements and conclusions. The Discussion chapter is very modest. Two literature items are referred to, including one already cited. The Conclusion chapter is a mixture of discussion and conclusion.

General and specific comments and suggestions
Comment 1: Page 1; “In Norway, sustainable crop yield production depends on the agro-climatic conditions, the persistence of

rainfall, soil quality, and other infrastructural development”. Please add reference to this sentence
Author Reply: As per the reviewer's suggestion, the authors have cited the sentence with the relevant citation numbered [1].

Comment 2: Page 1; “As the global population has been increasing, it is a significant challenge for the farmers to produce increased quantities and better quality grains.” Please add reference to this sentence
Author Reply: As per the reviewer's suggestion, the authors have cited the sentence with the citation numbered [2].

Comment 3: Page 2; I propose radically shortening section 1.1 or removing it. Information about the project can be found in the article, but it is too detailed.
Author Reply: The authors have removed the section 1.1 i.e. KORNMO Research Project. To provide generic information regarding the KORNMO project to the reader, we included the link to the KORNMO project in the footnote. Kindly please review this change on Page 2.

Comment 4: Page 2; “We identified that the weather data is helpful to predict farm-scale yield production, and the satellite data is beneficial to predict regional-scale yield production.” Please add reference to this sentence
Author Reply: As per the reviewer's suggestion, the authors have included the relevant citations to the sentence, numbered from 4 to 7 in the article.

Comment 5: Chapter 1.2, The hypotheses presented for objects other than in Norway have already been studied. The state of knowledge in this area would need to be added.
Author Reply: We agree with the reviewer that crop yield predictions from satellite images have been studied. Section 1.2 addresses the state-of-the-art knowledge on crop yield prediction of images and particular yield prediction from satellite images, so-called remote sensing. We have made the connection more apparent.

Comment 6: Chapters 2 and 3 are part review not methodology. The authors themselves write about this: „In Section 2, we report the history of Norwegian agriculture and grain production. In addition, we report the relevant studies of using remote sensing data and building deep learning models in agriculture improvements. In section 3, we define the current state-of-the-art approaches for the crop yield predictions using remotely sensed data”

I suggest moving it to the Introduction chapter.
Author Reply: As per the suggestion, we have moved the section 3 as part of Introduction section and made it a subsection i.e., section 1.2. We have made Related Work section as Section 2, to report the history of Norwegian agriculture and grain production along with basic deep learning models in agriculture improvements. In section 3 i.e., materials and methods, we explained the research methodology in detail.

Comment 7: Page 4; “Plant growth and wellness are affected by elements from its surroundings. According to Oregon State University, the four main environmental factors affecting plants growth are: light, temperature, water, and nutrition”. Please provide references for this sentence and subsequent sentences, but references for peer-reviewed material.
Author Reply: The authors have provided the peer-reviewed citations numbered from 31 to 38, for section 2.2 on Page 6 and 7.

Comment 8: Chapters 2.3, 2.3.1, 2.3.2 contain very basic knowledge. It proposes to shorten them.
Author Reply: The authors have revised the section 2.3, 2.3.1, and 2.3.2 as per the suggestion and shortened them with the most relevant information on Page 7 and 8.

Comment 9: I propose to remove the elaborate structure of sections 4.1; 4.2; 4.4 in which points are used.
Author Reply: The authors have removed the elaborated structure of the asked sections. In addition, authors have revised this section carefully and made it as a subsection of Section 3 i.e., Materials and Methods to make it more clearer for the reader. Please see the changes in section 3.1 in the article.

Comment 10: Page 11; “The weather is one of the main external factors that are crucial for farming. Grain farmers depend on periods with little precipitation in the spring so that the fields are dry enough to support heavy equipment for ploughing, harrowing, and sowing. After sowing, the temperature must be stable so that the seedlings sprout and precipitation throughout the summer is required to water the plants”. Please add references to those sentences

Author Reply: The authors have included the citations numbered from 54 to 56 as per the suggestion.

Comment 11: Page 12; “The use of satellite images is the current state-of-the-art for farm-scale crop yield prediction without intrusive and labour-intensive monitoring. Therefore, it is also a significant focus of this research study”. Please add references to those sentences
Author Reply: The authors have cited the current state-of-the-art studies numbered 8, 9, and 22 to the suggested sentences.

Comment 12: Page 13; “Image masking makes it possible to focus on portions of an image that are of interest. Masks can be applied so that it either highlights certain parts or removes irrelevant parts from the image. In the context of this research study, provided we know where the cultivated fields are located, masks can be used on the satellite images to remove everything not registered as a cultivated field or as extra information to highlight where the fields are”. Please add reference to this sentences Author Reply: The authors have included the references numbered 57 and 58 for the asked sentences.

Comment 13: “In conclusion, we acquire” – Avoid "We", "I" statements. Use the passive voice. «In this section, we present the research methodology we have followed for this research study”. Unnecessary sentence. After all, that is what the chapter is called. I suggest deleting it.
Author Reply: Resolved as suggested

Comment 14: Page 14; “Because the memory requirements of the complete dataset are too large to fit in GPU memory or even RAM, images are continuously read from storage”. Why hasn't the computing power of cloud services been used?
Author Reply: The respective reviewer is correct that the authors could have run the experiments on a cloud service. The research was done on the universities Nvidia DGX-2 cluster. A cloud service would still have the same limitation on loaded images to GPU or RAM.

Comment 15: General; The use of the unit of area is incorrect. Decares are only used in one place in the world, Norway. I understand the authors' preference, but it is necessary to change it to an area unit according to an international standard, the SI system ( m2, hm2, or ha).
https://www.bipm.org/en/publications/si-brochure

Author Reply: Resolved as suggested with the m2 SI unit system i.e., 1 decare = 1000 m2

Reviewer 2 Report

The authors in this paper proposed a neural networks based method for crop yield prediction at the farm-level based on data such as Sentinel-2 satellite images, weather data, farm data, and grain delivery data.
Overall, the paper is interesting and well-written.
 Here are my comments:

1) Section 3 does not reviews recent deep learning based models for crop yield prediction such as :
a) Simultaneous corn and soybean yield prediction from remote sensing data using deep transfer learning

b) A cnn-rnn framework for crop yield prediction

c) Deep gaussian process for crop yield prediction based on remote sensing data
2) Authors did not specify the exact architecture of the CNN part of the model? I think a detailed table is needed.

3) Did you feed the image patches directly to CNN? or you fed the handcrafted features such as NDVI to the 1D-CNNs?

Author Response

Reviewer 2

Comments and Suggestions from Reviewer
Recommendation: The authors in this paper proposed a neural networks-based method for crop yield prediction at the farm-level based on data such as Sentinel-2 satellite images, weather data, farm data, and grain delivery data. Overall, the paper is interesting and well-written.

Here are my comments:

Comment 1: Section 3 does not reviews recent deep learning based models for crop yield prediction such as :

  • Simultaneous corn and soybean yield prediction from remote sensing data using deep transfer learning

  • A cnn-rnn framework for crop yield prediction

  • Deep gaussian process for crop yield prediction based on remote sensing data

    Author Reply: Resolved as suggested by the reviewer. Please review changes in Section 1.2.3 on Page 5.

    Comment 2: Authors did not specify the exact architecture of the CNN part of the model? I think a detailed table is needed.
    Author Reply: The reviewer is correct and the authors revised the sections 3.4.1, 3.4.2 and 3.5.1. To make the model more clearer and understandable, authors included the figure 4 on Page 15, to explain the architecture used to implement the proposed CNN part of the model.

    Comment 3: Did you feed the image patches directly to CNN? or you fed the handcrafted features such as NDVI to the 1D-CNNs?
    Author Reply: We used handcrafted features in building LSTM model explained in section 3.5.1 and the model is tested with three sets of sequential inputs: Weather, vegetation indices, and a combination of weather and vegetation indices. We found (and explained in Section 4.2.3) that weather alone only results in an MAE of 93.63 kg/1000 m2, while vegetation indices see an improved MAE of 83.01 kg/1000 m2. With both weather and vegetation indices, the MAE improves to 82.29 kg/1000 m2.

    On the other hand, we fed satellite image patches directly to CNN and weather data to 1D-CNNs, explained in section 3.5.3. The results clearly shows that our novel approach of hybrid CNN model outperformed the LSTM model with the MAE of 86.69 kg/1000 m2 if we do not apply image masking; 76.57/1000 m2 MAE if we add image mask as a channel; and 76.27/ 1000 m2 MAE if we multiply the mask by each image channel, please review Section 4.4 for the detailed explanation of the results.

Round 2

Reviewer 1 Report

The authors responded to all comments and have corrected most of the shortcomings.